# Detection of human disease conditions by single-cell morpho-rheological phenotyping of blood

Nicole Toepfner[1,2,3], Christoph Herold[1,4], Oliver Otto[1,4,5], Philipp Rosendahl[1,4], Angela Jacobi[1], Martin Kräter[6], Julia Stächele[3], Leonhard Menschner[3], Maik Herbig[1], Laura Ciuffreda[7], Lisa Ranford-Cartwright[7], Michal Grzybek[8,9], Ünal Coskun[8,9], Elisabeth Reithuber[10,11], Geneviève Garriss[10,11], Peter Mellroth[10,11], Birgitta Henriques-Normark[10,11], Nicola Tregay[2], Meinolf Suttorp[3], Martin Bornhäuser[6], Edwin R Chilvers[2], Reinhard Berner[3], Jochen Guck[1]*

[1]Center of Molecular and Cellular Bioengineering, Biotechnology Center, Technische Universität Dresden, Dresden, Germany; [2]Department of Medicine, University of Cambridge, Cambridge, United Kingdom; [3]Department of Pediatrics, University Clinic Carl Gustav Carus, Technische Universität Dresden, Dresden, Germany; [4]Zellmechanik Dresden GmbH, Dresden, Germany; [5]ZIK HIKE, Universität Greifswald, Greifswald, Germany; [6]Department of Hematology and Oncology, University Clinic Carl Gustav Carus, Technische Universität Dresden, Dresden, Germany; [7]Institute of Infection, Immunity and Inflammation, University of Glasgow, Glasgow, United Kingdom; [8]Paul Langerhans Institute Dresden of the Helmholtz Centre Munich, University Hospital and Faculty of Medicine Carl Gustav Carus, Technische Universität Dresden, Dresden, Germany; [9]German Center for Diabetes Research, Neuherberg, Germany; [10]Department of Microbiology, Tumor and Cell Biology, Karolinska Institutet, Stockholm, Sweden; [11]Department of Clinical Microbiology, Karolinska University Hospital, Stockholm, Sweden

*For correspondence:
jochen.guck@tu-dresden.de

**Abstract** Blood is arguably the most important bodily fluid and its analysis provides crucial health status information. A first routine measure to narrow down diagnosis in clinical practice is the differential blood count, determining the frequency of all major blood cells. What is lacking to advance initial blood diagnostics is an unbiased and quick functional assessment of blood that can narrow down the diagnosis and generate specific hypotheses. To address this need, we introduce the continuous, cell-by-cell morpho-rheological (MORE) analysis of diluted whole blood, without labeling, enrichment or separation, at rates of 1000 cells/sec. In a drop of blood we can identify all major blood cells and characterize their pathological changes in several disease conditions in vitro and in patient samples. This approach takes previous results of mechanical studies on specifically isolated blood cells to the level of application directly in blood and adds a functional dimension to conventional blood analysis.

DOI: https://doi.org/10.7554/eLife.29213.001

## Introduction

Blood is responsible for the distribution of oxygen and nutrients, and centrally involved in the immune response. Consequently, its analysis yields crucial information about the health status of patients. The complete blood count, the analysis of presence and frequency of all major blood cells,

**eLife digest** When you are sick and go to the doctor, it is often not obvious what exactly is wrong — what is causing fever, nausea, shortness of breath or other symptoms. It is important to find this out quickly so that the right action can be taken. One of the first steps is to obtain a blood sample and to count how many of the different blood cells are present in it. This is called a complete blood count, and the information it provides has turned out to be surprisingly useful. A large number of certain white blood cells, for example, can show that the body is fighting an infection. But there might be several reasons why the number of white blood cells has increased, so this information alone is often not enough for a specific diagnosis.

There are many hundreds of possible tests that can supplement the results of a complete blood count. These might identify bacteria or measure the concentrations of certain molecules in the blood, for example. But which test will give the important clue that reveals the source of the illness? This can be difficult to predict. Although each test helps to narrow down the final diagnosis they become increasingly expensive and time-consuming to perform, and rapid action is often important when treating a disease.

Can we get more decisive information from the initial blood test by measuring other properties of the blood cells? Toepfner et al. now show that this is possible using a technique called real-time deformability cytometry. This method forces the blood cells in a small drop of blood to flow extremely rapidly through a narrow microfluidic channel while they are imaged by a fast camera. A computer algorithm can then analyze the size and stiffness of the blood cells in real-time. Toepfner et al. show that this approach can detect characteristic changes that affect blood cells as a result of malaria, spherocytosis, bacterial and viral infections, and leukemia. Furthermore, many thousands of blood cells can be measured in a few minutes — fast enough to be suitable as a diagnostic test.

These proof-of-concept findings can now be used to develop actual diagnostic tests for a wide range of blood-related diseases. The approach could also be used to test which of several drugs is working to treat a certain medical condition, and to monitor whether the treatment is progressing as planned.

DOI: https://doi.org/10.7554/eLife.29213.002

constitutes a basic, routine measure in clinical practice. It is often accompanied by analysis of blood biochemistry and molecular markers reflecting the current focus on molecular considerations in biology and biomedicine.

An orthogonal approach could be seen in the study of the overall rheological properties of blood. It is evident that the flow of blood throughout the body will be determined by its physical properties in the vasculature, and their alterations could cause or reflect pathological conditions (*Lichtman, 1973*; *Baskurt and Meiselman, 2003*; *Popel and Johnson, 2005*). In this context, blood is a poly-disperse suspension of colloids with different deformability and the flow properties of such non-Newtonian fluids have been the center of study in hydrodynamics and colloidal physics (*Lan and Khismatullin, 2012*). Probably due to the dominant importance of erythrocytes, at the expense of sensitivity to leukocyte properties, whole blood rheology has not resulted in wide-spread diagnostic application.

Focusing on the physical properties of individual blood cells has suggested a third possibility to glean maximum diagnostic information from blood. Various cell mechanics measurement techniques, such as atomic force microscopy (*Worthen et al., 1989*; *Rosenbluth et al., 2006*; *Lam et al., 2008*), micropipette aspiration (*Lichtman, 1973*; *Lichtman, 1970*; *Dombret et al., 1995*; *Ravetto et al., 2014*) or optical traps (*Lautenschläger et al., 2009*; *Ekpenyong et al., 2012*; *Paul et al., 2013*), have been used to show that leukocyte activation, leukemia, and malaria infection, amongst many other physiological and pathological changes, lead to readily quantifiable mechanical alterations in the major blood cells (*Worthen et al., 1989*; *Lautenschläger et al., 2009*; *Schmid-Schönbein et al., 1973*; *Suresh et al., 2005*; *Rosenbluth et al., 2008*; *Bow et al., 2011*; *Gossett et al., 2012*). These proof-of-concept studies have so far been done on few tens of specifically isolated cells. This line of research has not progressed towards clinical application for lack of an appropriate measurement technique that can assess single-cell properties of sufficient number directly in blood.

This report aims to close this gap by presenting a novel approach for high-throughput single-cell morpho-rheological (MORE) characterization of all major blood cells in continuous flow. Mimicking capillary flow, we analyze human blood without any labeling or separation at rates of 1000 cells/sec. We show that we can sensitively detect morphological and rheological changes of erythrocytes in spherocytosis and malaria infection, of leukocytes in viral and bacterial infection, and of malignant transformed cells in myeloid and lymphatic leukemias. Readily available quantitative morphological parameters such as cell shape, size, aggregation, and brightness, as well as rheological information of each blood cell type with excellent statistics might not only inform further investigation of blood as a complex fluid. It also connects many previous reports of mechanical changes of specifically isolated cells to a measurement now done directly in blood. As such, it adds a new functional dimension to conventional blood analysis — a MORE complete blood count — and, thus, opens the door to a new era of exploration in investigating and diagnosing hematological and systemic disorders.

## Results

### Establishment of morpho-rheological analysis

In order to establish the normal MORE phenotype of cells found in blood, we obtained venous, citrate-anticoagulated blood of healthy donors, of which 50 µl was diluted in 950 µl of measurement buffer with a controlled elevated viscosity, but without any additional labeling, sorting, or enrichment. The cell suspension was then pumped through a micro-channel not unlike micro-capillaries in the blood vasculature (*Figure 1A*). Brightfield images of the cells, deformed by hydrodynamic shear stresses in the channel (*Mietke et al., 2015*), were obtained continuously by RT-DC (*Otto et al., 2015*) (see Materials and methods; *Video 1*). These images revealed distinct differences in overall morphology, brightness, and amount of deformation between all major cell types found in blood (*Figure 1B*). RT-DC further enabled the continuous, real-time quantification of the cross-sectional area and of the deformed shape (see detailed description in Materials and methods and *Figure 1—figure supplement 1*) of an, in principle, unlimited number of cells at measurement rates of 100–1000 cells/sec (*Figure 1C*). For each cell detected and analyzed, an image was saved and the average pixel brightness within the cell determined (*Figure 1D*, *Figure 1—figure supplement 1*). This single-cell MORE analysis of blood revealed distinct and well-separated cell populations in the space spanned by the three parameters (*Video 2*). Notably, size and brightness alone — parameters not unlike those accessible by light scattering analysis in standard flow cytometers — were sufficient for the identification of the cell types (*Figure 1D*), so that deformation, as an additional and independent parameter, was available for assessing their functional changes. The identity of the individual cell populations by size and brightness was established by magnetic cell sorting, controlled by fluorescence immunophenotyping, and subsequent MORE analysis (*Figure 1—figure supplement 2*). A key feature is the very clear separation of the abundant erythrocytes (red blood cells; RBCs) from other cells as a result of their much greater deformation and lower brightness. This feature gives access to leukocyte properties directly in diluted whole blood, without the potentially detrimental effects of hemolysis (see *Figure 1—figure supplement 3*) or other separation steps, which are required for analysis with cell mechanics techniques with lower specificity and throughput, or non-continuous measurement. This aspect contributes to the well-established field of hemorheology the possibility to interrogate mechanical properties of all individual blood cells and to specifically investigate their contribution to the overall blood rheological properties.

In extensive tests of the variability of this approach, MORE phenotyping yielded identical results in repeated measurements of blood from the same donor, with sodium citrate added as an anticoagulant and for different storage times (*Figure 1—figure supplement 4*), between different donors of both sexes (*Figure 1—figure supplement 5*), and blood samples taken at different times during the day (*Figure 1—figure supplement 6*). This robustness served to establish a norm for the different cell types (*Figure 1E*). MORE analysis provided the identity and frequency of all major white blood cells as with a conventional differential blood count (*Figure 1F*; *Supplementary file 1*) — obtained from a single drop of blood, with minimal preparation, and within 15 min. Going beyond this current gold-standard of routine blood cell analysis, and importantly also beyond all other single-blood-cell mechanical analysis studies to date, MORE phenotyping allowed the sensitive characterization of pathophysiological changes of individual cells directly in merely diluted whole blood. In

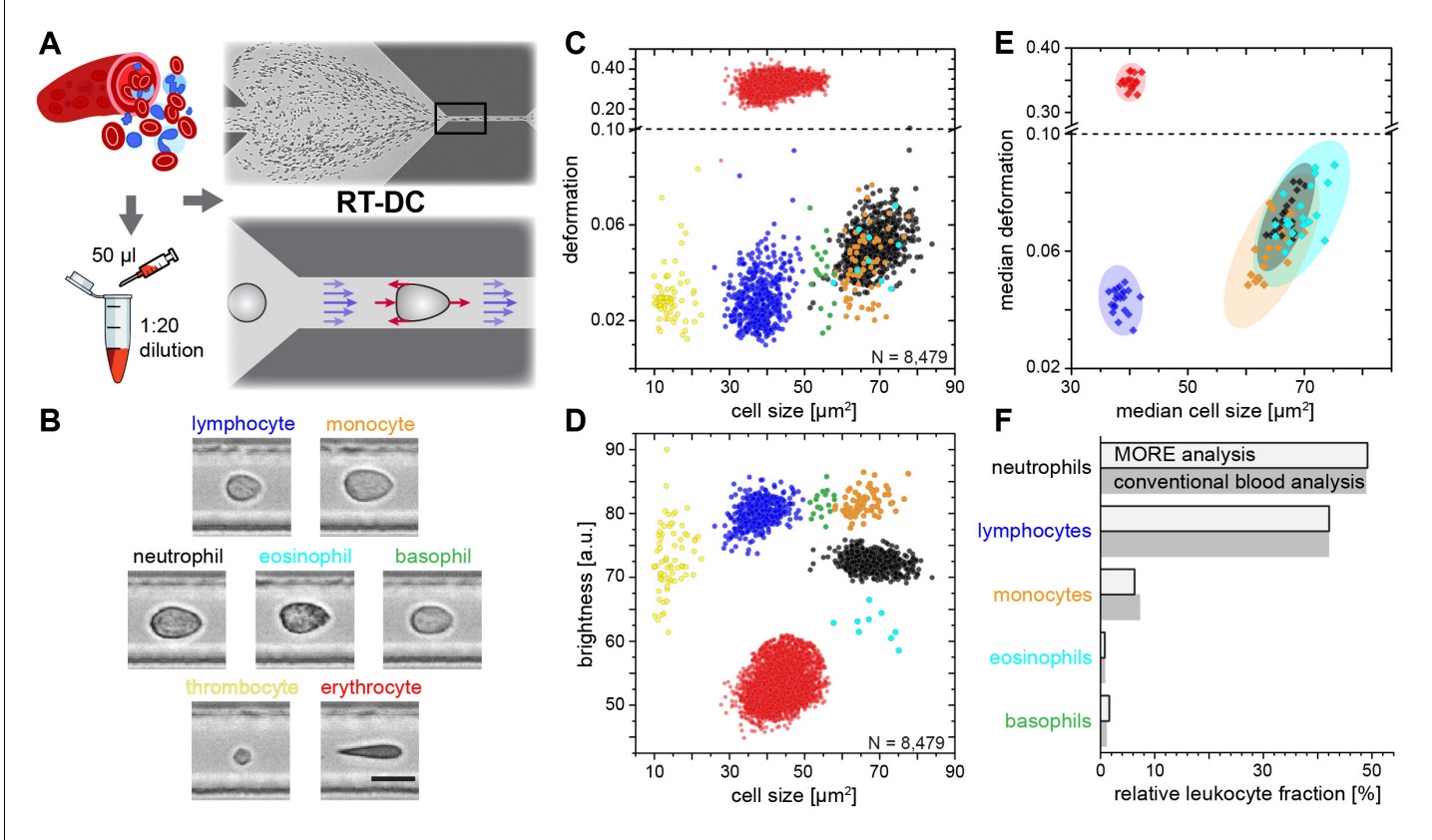

**Figure 1.** Single-cell, morpho-rheological phenotyping of blood. (**A**) Analysis of whole, diluted blood. Hydrodynamic shear forces (red arrows) induce deformation of cells passing through a microfluidic channel ($20 \times 20$ μm$^2$) at speeds of more than 30 cm/s (blue arrows). (**B**) Representative images of blood cell types acquired. Scale bar is 10 μm. Images are analyzed for cell size as well as (**C**) cell deformation and (**D**) average cell brightness. Each dot represents one of $N$ measurement events. (**E**) Normal range of deformation and size of cell populations from healthy donors. Each diamond represents the median of one donor; transparent ellipses indicate 95% confidence areas. (**F**) Comparison of MORE cell counts with conventional blood count.

DOI: https://doi.org/10.7554/eLife.29213.003

The following source data and figure supplements are available for figure 1:

**Source data 1.** Source data for Figure 1.

DOI: https://doi.org/10.7554/eLife.29213.010

**Figure supplement 1.** Definition of RT-DC parameters and illustration of gates.

DOI: https://doi.org/10.7554/eLife.29213.004

**Figure supplement 2.** Brightness and cell size of purified leukocyte subpopulations in MORE analysis.

DOI: https://doi.org/10.7554/eLife.29213.005

**Figure supplement 3.** Effect of red blood cell lysis on morpho-rheological properties of leukocytes.

DOI: https://doi.org/10.7554/eLife.29213.006

**Figure supplement 4.** Stability of results with anti-coagulant and storage time.

DOI: https://doi.org/10.7554/eLife.29213.007

**Figure supplement 5.** Inter-donor variation of deformation and cell size.

DOI: https://doi.org/10.7554/eLife.29213.008

**Figure supplement 6.** Intra-day variation of deformation and cell size.

DOI: https://doi.org/10.7554/eLife.29213.009

the following, we exemplarily demonstrate, in turn for each of the blood cell types, the new possibilities of gaining MORE information from an initial blood test as a time-critical step in generating specific hypotheses and steering further investigation enabled by this approach.

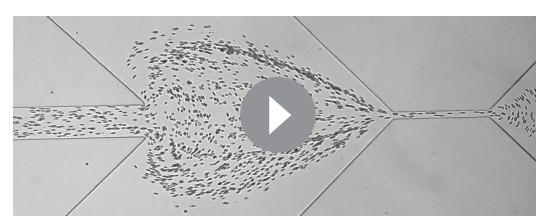

**Video 1.** RT-DC in action. Video of the microfluidic channel system during RT-DC measurement of diluted whole blood. The cell suspension flows from left to right through the channel. Cells enter on the left and are focused by sheath flow from the top and bottom of the frame towards the narrow RT-DC measurement channel of 300 μm length and 20 μm width and height in the right half of the image. RT-DC measurements are carried out on the cells that travel through the last third of the length of the measurement channel.
DOI: https://doi.org/10.7554/eLife.29213.011

## Detection of morpho-rheological changes in erythrocytes

Spherocytosis is a prototypical hereditary disease in humans in which genetic changes (here ankyrin and spectrin mutations) cause abnormal shape and mechanical properties of erythrocytes. Current diagnosis is based on the detection of abnormal cell shapes in a blood smear, followed up by assessment of the osmotic fragility quantified by Acidified Glycerol Lysis Time (AGLT) or by osmotic gradient ektacytometry. These manual assays take time and do not lend themselves to quick, initial screening. MORE analysis of the blood of patients with spherocytosis directly revealed significantly less deformed and smaller erythrocytes than normal (*Figure 2A–C*) as the functional correlate of the cytoskeletal mutation. The differences are so clear (*Figure 2—figure supplement 1*) that this analysis can serve as a fast primary and cheap screening test for spherocytosis. Detection of such RBC changes would then warrant confirmation by more specific analysis using flow-cytometric detection of Eosin-5-Maleimide staining (EMA test) or the direct detection of the mutation by PCR, which require specific preparation, are more expensive, and thus benefit from a strong and clear initial hypothesis.

A change in RBC deformability has also been implicated in malaria pathogenesis, since single cells infected by parasites have been reported to be stiffer (*Bow et al., 2011*). This insight has not progressed towards clinical application and the gold standard in malaria diagnosis is still a manual thick blood smear analysis. To evaluate whether MORE analysis could provide a sensitive, automated alternative, we analyzed populations of RBCs exposed in vitro to *Plasmodium falciparum* (*P.f.*) with a parasitemia (percentage of actually infected cells) of 7–8% at time points over the 2 day parasite life cycle. We found a clear, significant, and increasing reduction in the deformation of the entire exposed RBC population detectable after 4 hr (*Figure 2D–F*; *Figure 2—figure supplement 2*). Inspection of the individual cell images revealed the appearance of characteristic features likely associated with the maturation of parasites inside a subset of RBCs (*Figure 2D,E* insets; *Figure 2—figure supplement 2*). These features permitted the direct identification of positively infected cells, whose relative frequency peaked at 36 hr (*Figure 2—figure supplement 2*). The separate assessment of overtly infected cells showed an even greater reduction in deformation than observed in the entire exposed population (*Figure 2F*; *Figure 2—figure supplement 2*), which — extrapolating our in vitro results to the situation in vivo — relates to the possibility of clearance of stiff, infected cells from the circulation by the spleen (*Cranston et al., 1984*; *Shelby et al., 2003*). However, this small fraction of stiffer cells alone cannot account for the reduced deformation of the whole population, so that a bystander stiffening of exposed but non-infected cells seems involved (*Paul et al., 2013*).

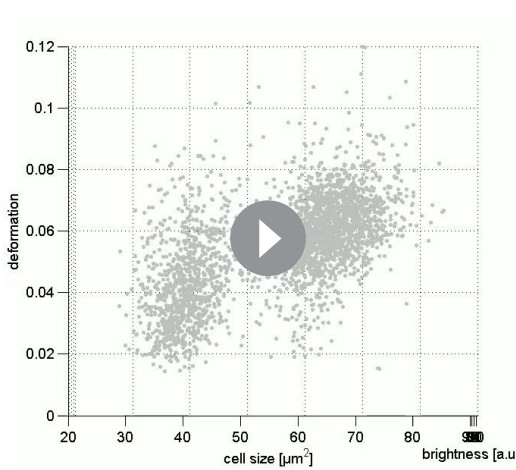

**Video 2.** 3D visualization of the separation of leukocyte populations. Rotating angle view in the space of deformation, cell size and cell brightness. Cell identification in order of appearance by coloring: lymphocytes (blue), neutrophil granulocytes (black), eosinophil granulocytes (cyan), monocytes (orange), basophil granulocytes (green).
DOI: https://doi.org/10.7554/eLife.29213.012

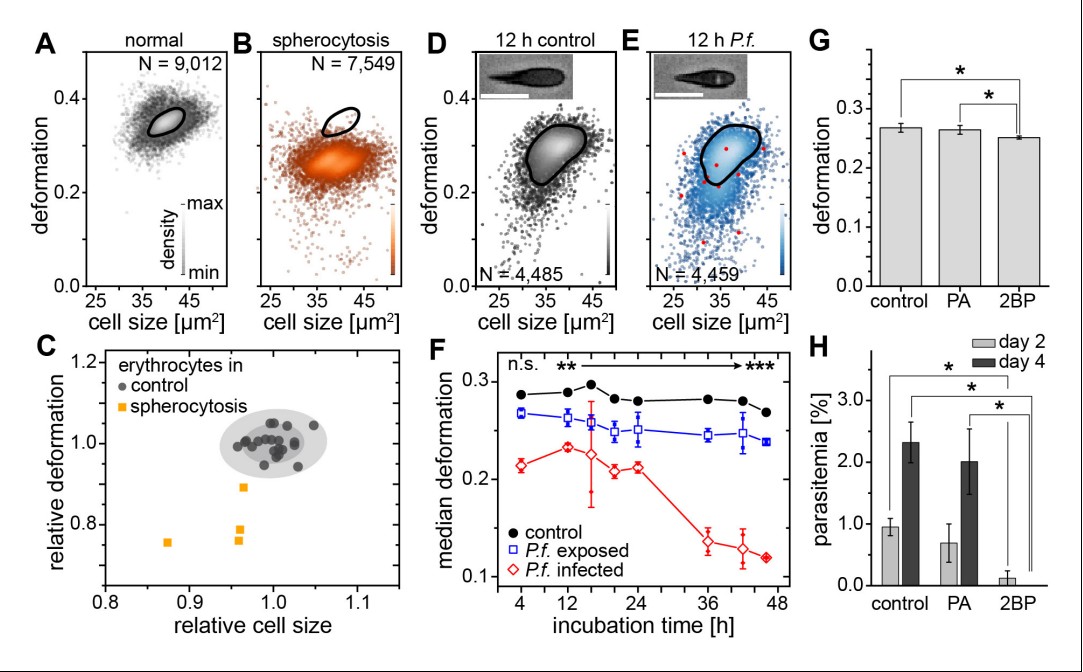

**Figure 2.** Detection of RBC pathologies — spherocytosis and malaria. Exemplary density plots of RBC size vs. deformation in samples from (**A**) healthy donor and (**B**) patient with spherocytosis. (**C**) Relative median RBC deformation and size in patients with spherocytosis (orange, *n* = 4 patients) compared to controls (black, *n* = 21 donors as in *Figure 1E* with 68% and 95% confidence ellipses). Density plots of size vs. deformation of (**D**) control RBCs and (**E**) RBCs exposed to *P.f.* (blue), both after 12 hr incubation. Insets show cells with and without image feature (white inclusion), positively indicating actual parasite infection. Infected RBCs are indicated by red dots. Scale bars, 10 μm. (**F**) Evolution of RBC deformation over 46 hr time course of control (black), *P.f.* exposed (blue) and *P.f.* infected RBCs (red); open squares and diamonds, mean ±SD, *n* = 2; filled squares, individual medians, **p<0.01, ***p<0.001. (**G**) Reduced deformation of 2BP-treated RBCs compared to PA- and non-treated controls (mean ±SD of population medians, *n* = 4 donors, *p<0.05). (**H**) Reduced parasitemia in 2BP- compared to PA- and non-treated controls at 2 and 4 days post infection. Error bars: SD binomial, *p<0.0125.

DOI: https://doi.org/10.7554/eLife.29213.013

The following source data and figure supplements are available for figure 2:

**Source data 1.** Source data for Figure 2.
DOI: https://doi.org/10.7554/eLife.29213.016
**Figure supplement 1.** Comparison of erythrocytes in spherocytosis with the healthy control.
DOI: https://doi.org/10.7554/eLife.29213.014
**Figure supplement 2.** In vitro infection of erythrocytes with *Plasmodium falciparum*.
DOI: https://doi.org/10.7554/eLife.29213.015

Reduced membrane-cytoskeleton interactions have previously been correlated with elliptocytic RBCs and resistance to *P.f.* infection (*Chishti et al., 1996*). The characteristic biconcave morphology of RBCs can be chemically altered by the use of 2-bromo-palmitate (2 BP), an efficient inhibitor of palmitoyl acyltransferases (*Biernatowska et al., 2013*). Here, 2 BP-treated RBC samples showed changes in deformation (*Figure 2G*) with a concurrent reduction in *P.f.* infectivity (*Figure 2H*), compared to buffer control or RBCs treated with palmitic acid (PA). PA is an analogue of 2 BP that does not inhibit palmitoylation (*Biernatowska et al., 2013*). Since both, 2 BP and PA readily accumulate in the membranes, but only 2 BP causes a reduction in infectivity of *P.f.*, we suggest that palmitoylation of RBC proteins is important for RBC morphology and infectivity of *P.f.* While a previous report had found no change in infectability of RBCs treated with 2 BP (*Jones et al., 2012*), the difference could stem from the different RBC receptors involved in invasion by the different parasite clones (3D7 vs. HB3), which in turn are differentially affected by palmitoylation. Thus, MORE analysis has the potential not only to simplify, automate, and speed up malaria diagnosis, but also to provide additional quantitative information aiding research on the pathogenesis of the disease (*Koch et al., 2017*).

## Detection of morpho-rheological changes in leukocytes

While RBC mechanics has already been used for clinical diagnostics using rheoscopes and ektacytometers for over 40 years (*Schmid-Schönbein et al., 1973*; *Cranston et al., 1984*; *Reid et al., 1976*), leukocyte mechanics has not been utilized for diagnostic purposes. This is likely due to their increased stiffness compared to RBCs and a lack of convenient techniques capable of sufficiently deforming them in suspension — their physiological state. Until recently, techniques with sufficient throughput, obviating the need for specifically isolating the relevant cells of interest, which always bears the potential of inadvertent cell change (see *Figure 1—figure supplement 3*), did not exist. In this sense, the mechanical phenotyping of diagnostic changes of leukocytes directly in diluted whole blood is the most transformative application area of MORE analysis. For example, there have been proof-of-concept studies on the mechanical changes associated with activation of isolated neutrophils showing a stiffening, in line with the pronounced actin cortex that is a hallmark of neutrophil activation (*Worthen et al., 1989*; *Rosenbluth et al., 2008*). MORE analysis of the in vitro neutrophil activation in blood with the bacterial wall-derived tripeptide fMLP confirmed that neutrophils were indeed less deformed and smaller within the first 15 min post fMLP treatment. Interestingly, the subsequent time-course showed a reversal to more deformed and larger cells (*Figure 3A,B*; *Figure 3—figure supplement 1*). These observations by themselves do not permit a conclusion about a change in cell stiffness, since a smaller size also leads to less stress acting on the cells in the channel, and less deformation (*Mietke et al., 2015*; *Mokbel et al., 2017*). Thus, we also calculated the apparent Young's modulus of the cells, which increased from $E = 742 \pm 12$ Pa to $E = 853 \pm 20$ Pa (mean ±SEM. p=0.009, $n = 5$) during the first 15 min, and then subsequently reverted to values statistically indistinguishable but slightly lower than before stimulation (15–30 min: $E = 717 \pm 9$ Pa, p=0.347; 30–45 min: $E = 719 \pm 7$ Pa, p=0.117; 45–60 min: $E = 731 \pm 11$ Pa, p=0.465). Such mechanical activation kinetics of neutrophils has not been reported before as the lower measurement rate of previous techniques yielded only cumulative data over the time period investigated.

We also found a similar increase in size and greater deformation of the neutrophils at the later time points in an experimental medicine trial, where healthy human volunteers inhaled lipopolysaccharide (LPS; from *E. coli*) (*Figure 3A,B*; *Figure 3—figure supplement 1*). Also, infecting blood in vitro with *Staphylococcus aureus* (*S. aureus*), a Gram-positive bacterium and one of the major pathogens responsible for life-threatening infections world-wide, resulted in larger and more deformed neutrophils, measured between 30–60 min after blood stimulation (*Figure 3A,B*; *Figure 3—figure supplement 2*).

Congruently, blood taken from patients with an acute lung injury (ALI) of most likely bacterial origin had larger and more deformed neutrophils compared to healthy controls (*Figure 3C,E,H*). The same neutrophil response was found in blood samples from patients hospitalized with viral respiratory tract infections (RTI; *Figure 3D,E,H*). Also monocytes responded by a size increase in both RTI and ALI patients and after in vitro stimulation with *S. aureus*, but showed a significantly increased deformation only in viral RTI, while blood lymphocytes did not show any consistent response (*Figure 3F–H*; *Figure 3—figure supplements 2* and *3*). The lymphocyte response changed when analyzing blood of patients with acute Epstein-Barr-virus (EBV) infection, which is known to also stimulate the lymphatic system, where both monocytes and lymphocytes showed an increase in cell size and deformation, while neutrophils showed less of a response (*Figure 3I–L*, *Figure 3—figure supplement 3*). These results suggest that MORE blood analysis might be sufficiently sensitive to distinguish bacterial from viral infections, and potentially other inflammatory diseases, by the differential response of selective blood leukocyte populations. This possibility will be followed up in future specific trials. Importantly, MORE blood analysis is of special interest for blood tests in neonatology with patients at high risk of infections but only minute amounts of blood available for diagnostics, or to characterize neutrophils in neutropenic patients, as it merely requires longer data acquisition periods.

## Detection of morpho-rheological changes in malignant transformed blood cells

Blood cancers, or leukemias, affecting both myeloid and lymphoid cell lineages, are a further large area where MORE analysis could potentially contribute fundamental insight, aid diagnosis, and improve therapy monitoring. While solid cancer cell mechanics has been a focus of cell mechanics

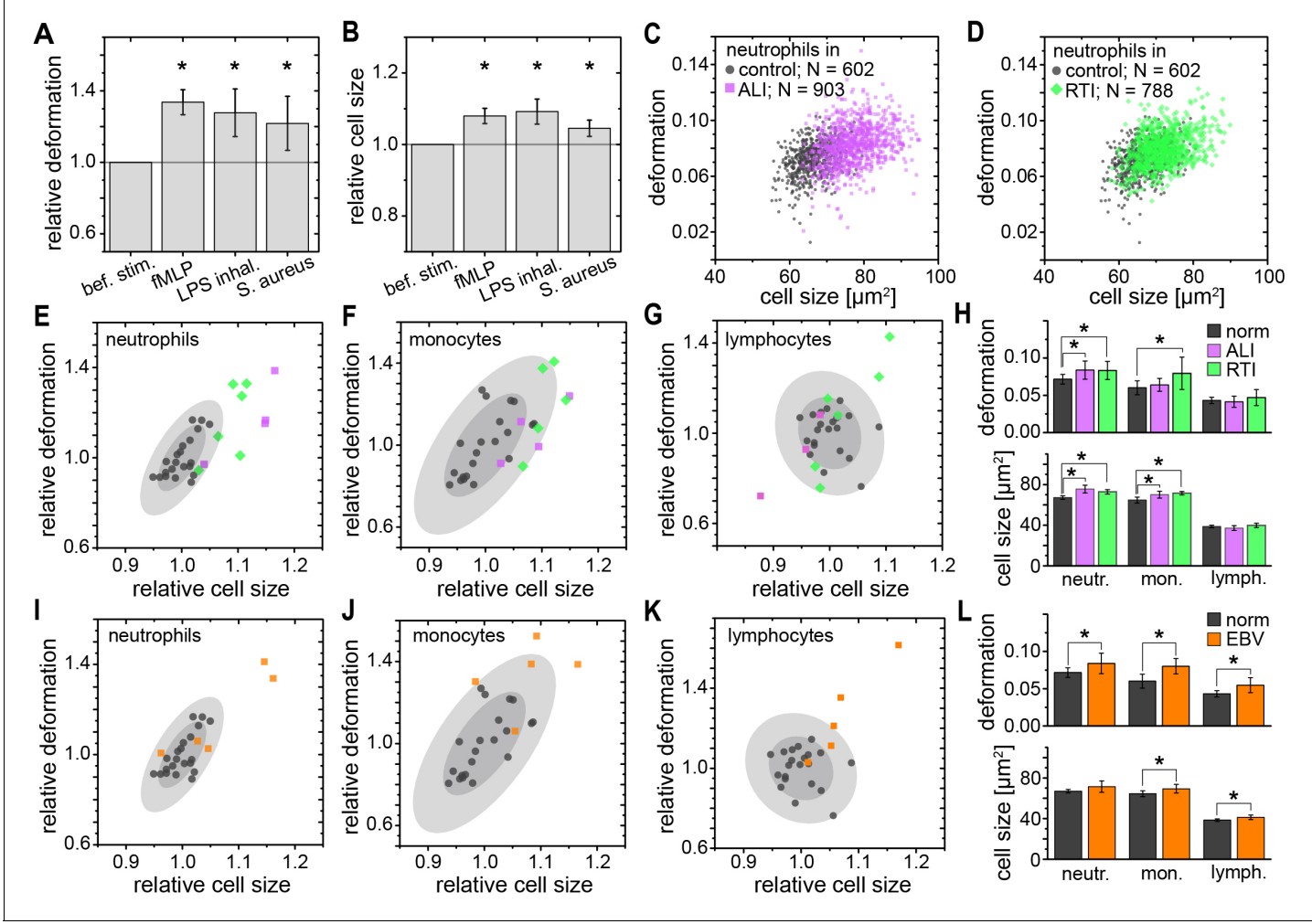

**Figure 3.** Identification of leukocyte activation and infection in vitro and in vivo. Relative change (mean ±SD) in (A) deformation and (B) size of neutrophils in diluted whole blood after fMLP (n = 5 donors; see *Figure 3—figure supplement 1*) and *S. aureus* (n = 4 donors; see *Figure 3—figure supplement 2*) stimulation in vitro measured 15–30 min and 30–60 min after stimulation, respectively, and LPS inhalation (n = 2 donors; see *Figure 3—figure supplement 1*) in vivo measured 135 min after inhalation. Exemplary scatter plots of size vs. deformation of neutrophils in blood of a patient with (C) ALI (magenta) and (D) RTI (green) compared to controls (black). Medians of size and deformation of (E, I) neutrophils, (F, J) monocytes, and (G, K) lymphocytes in blood samples of patients with E, F, G, ALI (n = 4 patients; magenta) and RTI (n = 6 patients; green), and I, J, K, EBV infection (n = 5 patients; orange) relative to the norm (black, n = 21 donors as in *Figure 1E* with 68% and 95% confidence ellipses). (H, I) Mean and SD of these results, *p<0.05. For typical scatter plots of size vs. deformation of all three cell types and all three disease conditions see *Figure 3—figure supplement 3*.

DOI: https://doi.org/10.7554/eLife.29213.017

The following source data and figure supplements are available for figure 3:

**Source data 1.** Source data for Figure 3.
DOI: https://doi.org/10.7554/eLife.29213.021

**Figure supplement 1.** Neutrophil response in blood during in vitro and in vivo stimulation.
DOI: https://doi.org/10.7554/eLife.29213.018

**Figure supplement 2.** In vitro stimulation of blood with *Staphylococcus aureus*.
DOI: https://doi.org/10.7554/eLife.29213.019

**Figure supplement 3.** Single cell distributions of neutrophils, lymphocytes, and monocytes from patients with ALI, RTI, and EBV compared to controls.
DOI: https://doi.org/10.7554/eLife.29213.020

research and extensively documented (*Suresh, 2007*; *Kumar and Weaver, 2009*; *Guck and Chilvers, 2013*), the mechanical properties of blood cancers are comparatively understudied.

The available research on mechanics of leukemic cells has been undertaken either on cell lines or fully purified cells (*Lichtman, 1973*; *Rosenbluth et al., 2006*; *Lam et al., 2008*; *Lichtman, 1970*;

*Dombret et al., 1995*; *Lautenschläger et al., 2009*; *Ekpenyong et al., 2012*; *Rosenbluth et al., 2008*; *Zheng et al., 2015*) but so far not directly in blood. MORE analysis of the blood of patients with acute myeloid (AML) and lymphatic leukemias (ALL) revealed the new presence of atypical cell populations — the characteristic immature blasts not normally present in healthy donors (*Figure 4A–C*). Cell populations gated for AML revealed less deformed cells but of about the same size compared to healthy and fully differentiated myeloid cells (*Figure 4D*, *Figure 4—figure supplement 1*), in line with previous results (*Rosenbluth et al., 2006*; *Lichtman, 1970*; *Dombret et al., 1995*; *Lautenschläger et al., 2009*; *Ekpenyong et al., 2012*). ALL blast cells were larger in size compared to mature lymphocytes, but did not show any consistent trend in deformation (*Figure 4D*; *Figure 4—figure supplement 1*). Since cell size and deformation in the channel are interrelated (*Mietke et al., 2015*; *Otto et al., 2015*; *Mokbel et al., 2017*), which can be seen by the isoelasticity lines parameterizing the deformation-size space (*Figure 4D*), we also calculated the apparent Young's modulus of these cells (*Figure 4—figure supplement 1*). Together these results show that mature lymphocytes, ALL blasts, mature myeloid cells, and AML blasts have increasing levels of stiffness, consistent with the composite findings of previous reports (*Lichtman, 1973*; *Rosenbluth et al.,*

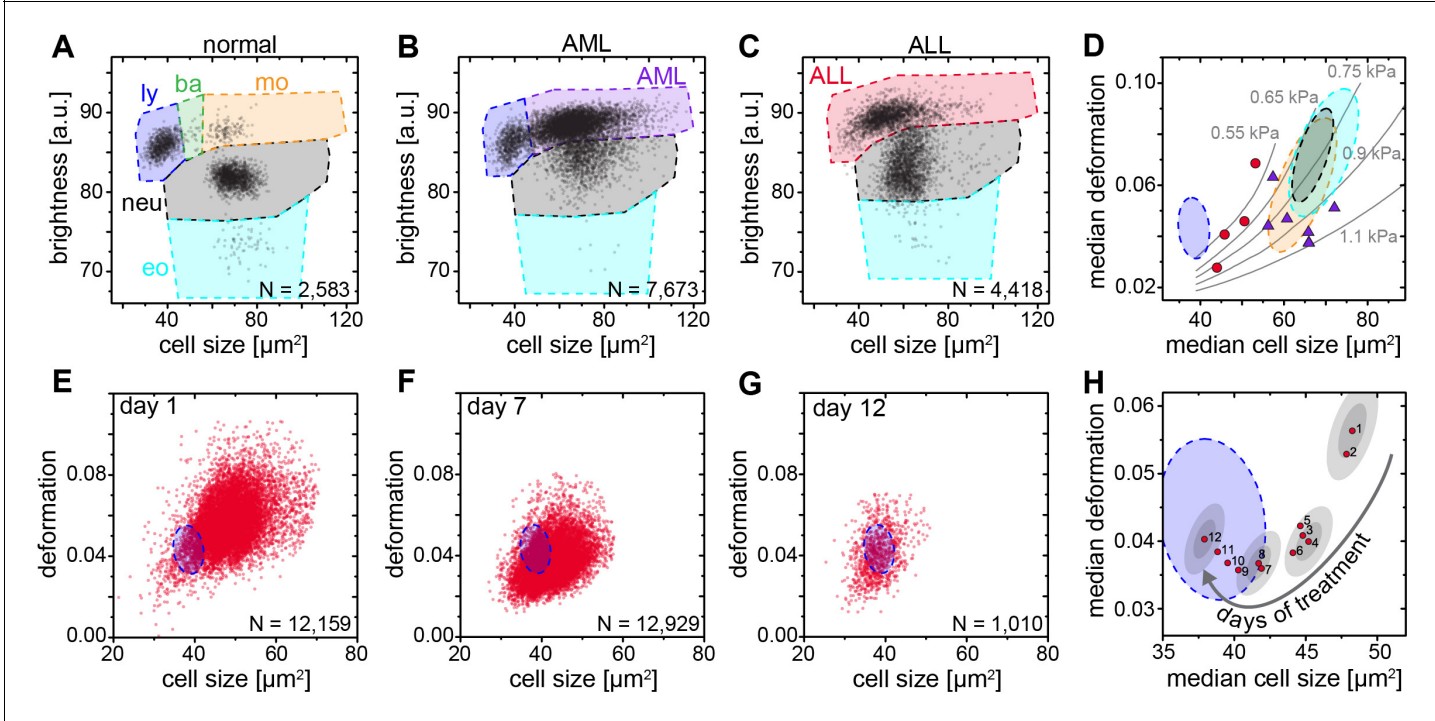

**Figure 4.** Detection and distinction of leukemia subtypes and monitoring of treatment effects. (**A**) Normal brightness vs. size scatter plot of a healthy donor with the gates (shaded areas) used to identify lymphocytes (ly), basophils (ba), monocytes (mo), neutrophils (neu) and eosinophils (eo). (**B**) Exemplary brightness vs. size scatter plot in AML; blast cells were found in (ba) and (mo) gates. (**C**) Exemplary brightness vs. size scatter plot in ALL; blast cells were found in (ly), (ba), and (mo) gates. (**D**) Medians of deformation and size for the respective gates in blood samples of ALL (red circles, n = 4 patients) and AML patients (purple triangles, n = 7 patients). Shaded areas in D (color as in A) represent 95% confidence ellipses of the respective cell type norm (n = 21 donors, as in *Figure 1E*). Gray lines represent lines of equal elasticity calculated for purely elastic objects. Scatter plots of ALL blast deformation and size at (**E**) one; (**F**) seven, and (**G**) twelve days post therapy start. Blue shaded areas in E-H represent 95% confidence ellipses of the lymphocyte norm (n = 21 donors, as in *Figure 1E*). (**H**) Median deformation and size of ALL cells during 12 days of treatment (red dots, days as numbers). Gray shaded areas surrounding data of days 1, 4, 8, and 12 represent the 68% (inner) and 95% (outer) confidence area of a single measurement (according to lymphocyte confidence in *Figure 1—figure supplement 6H*).

DOI: https://doi.org/10.7554/eLife.29213.022

The following source data and figure supplement are available for figure 4:

**Source data 1.** Source data for Figure 4.
DOI: https://doi.org/10.7554/eLife.29213.024
**Figure supplement 1.** Comparison of ALL and AML blast cells with the norm.
DOI: https://doi.org/10.7554/eLife.29213.023

*2006*; *Lam et al., 2008*; *Lichtman, 1970*; *Dombret et al., 1995*; *Lautenschläger et al., 2009*; *Ekpenyong et al., 2012*; *Rosenbluth et al., 2008*). This is quite different than the general trend in solid tumors, where cancer cells are found to be more deformable than their healthy counterparts (*Suresh, 2007*; *Kumar and Weaver, 2009*; *Guck and Chilvers, 2013*). Sensibly, the differential stiffness of AML and ALL blasts, and its potential further increase with chemotherapy, has been implicated in the occurrence of leukostasis (*Lam et al., 2008*; *Rosenbluth et al., 2008*; *Lam et al., 2007*). MORE analysis might not only permit screening for novel therapeutic targets to soften cells (*Gossett et al., 2012*; *Di Carlo, 2012*; *Surcel et al., 2015*), but also assessing the risk of leukostasis directly in each patient.

Finally, by following the ALL blast population in a patient over 12 days of methylprednisolone treatment we could monitor the return to the normal morpho-rheological fingerprint of blood (*Figure 4E–H*). The evolution of this fingerprint likely comprises multiple contributions with blast cells undergoing apoptosis over a time course of 2–7 days (*Ito et al., 1996*), which is associated with an increase in stiffness (*Lam et al., 2007*). Blast cells are sequestered by the spleen and new, but immature and likely stiffer, blast cells are being added to the circulation from the bone marrow. There could also be ALL subclones with different morpho-rheological characteristics that respond differently and at different times to treatment. And the final increase in deformation from day 9 to 12 coincides with the addition of cytostatic drugs (vincristine, daunorubicin) to the methylprednisolone treatment. Dissecting this multifaceted response will be aided by adding simultaneous fluorescence identification of the cells in the future (*Rosendahl, 2017*). Of note, of the conventional biomarkers and techniques that are used in the diagnosis of leukemia (see *Supplementary file 2*), only morphological analysis of air-dried Romanowsky-stained blood (or bone marrow) smears is traditionally applied to monitor treatment success in ALL. The response to treatment is one of the most powerful prognostic in vivo markers of leukemia survival. In pediatric ALL the number of blasts at day eight after start of methylprednisolone treatment is predictive of the relapse rate (<1000 blasts/µl of blood: relapse rate 20–30%; >1000 blasts/µl of blood: relapse rate 50–80%). MORE analysis provides at least the same information as conventional morphological analysis, but in a shorter amount of time and with smaller sample sizes required (for a comparison between MORE analysis and conventional biomarkers, see *Supplementary file 2*). In summary, MORE blood analysis can be used to monitor morpho-rheological effects of chemotherapy and the successful replacement of lymphoblasts with mature lymphocytes in a quantitative manner. This last finding also touches upon the study of hematopoietic differentiation of cells in the bone marrow, which is an obvious further potential area of application of this approach.

## Discussion

Morpho-rheological phenotyping allows individual blood cell mechanics to be studied in a range of human diseases and takes cell mechanical phenotyping to an entirely new level. While established techniques such as micropipette aspiration (*Lichtman, 1973*; *Lichtman, 1970*; *Dombret et al., 1995*; *Ravetto et al., 2014*), indentation by cell pokers and atomic force microscopes (*Worthen et al., 1989*; *Rosenbluth et al., 2006*; *Lam et al., 2008*), or optical trapping (*Lautenschläger et al., 2009*; *Ekpenyong et al., 2012*; *Paul et al., 2013*) have provided important proof-of-concept insight over the last decades, the recent advent of microfluidic techniques approaching the throughput of conventional flow cytometers (*Gossett et al., 2012*; *Otto et al., 2015*; *Zheng et al., 2015*; *Byun et al., 2013*; *Lange et al., 2015*) has finally brought mechanical phenotyping close to real-world applications (*Guck and Chilvers, 2013*; *Tse et al., 2013*). Amongst the latter techniques, RT-DC stands out because it can continuously monitor an, in principle, unlimited number of cells, which enables the direct sensitive assessment of the state of all major cell types found in blood. A volume as small as 10 µl can be analyzed cell-by-cell, with only dilution in measurement buffer to adjust cell density and prevent sedimentation, but no labeling, enrichment or separation, which could otherwise cause inadvertent activation of blood cells. The conventional blood count is extended by information about characteristic, and diagnostic, morpho-rheological changes of the major cell types. Cell mechanics and morphology are inherent and sensitive markers intimately linked to functional changes associated with the cytoskeleton (*Chimini and Chavrier, 2000*; *Fletcher and Mullins, 2010*; *Kasza et al., 2007*; *Patel et al., 2012*; *Salbreux et al., 2012*) and other intracellular shape-determining and load-bearing entities (*Rowat et al., 2006*; *Munder et al., 2016*).

Thus, label-free, disease-specific morpho-rheological blood signatures are a novel resource for generating hypotheses about the underlying molecular mechanisms. The availability of such parameters in real-time, easily combined with conventional fluorescence detection (*Rosendahl, 2017*), are the necessary prerequisite for future sorting of morpho-rheologically distinct subpopulations, which then provides a novel opportunity for further molecular biological analysis. Of course, at present, MORE phenotyping provides a sensitive, but not a very specific marker. For example, neutrophil softening could be a signature of different underlying pathological changes. In the future, fuller exploration of the large combinatorial space afforded by the multi-parametric response of the various blood cells, exploiting many additional morpho-rheological parameters in conjunction with machine learning, and inclusion of conventional fluorescence-based marker analysis (*Rosendahl, 2017*) will further increase the specificity of this approach. Apart from now enabling realistic blood cell research ex vivo close to physiological conditions, delivering for example previously unavailable information about leukocyte activation kinetics, and after further in-depth studies of the phenomena reported here, MORE phenotyping could have a tangible impact on diagnosis, prognosis, and monitoring of treatment success of many hematological diseases as well as inflammatory, infectious, and metabolic disorders. Beyond blood analysis, MORE phenotyping has the potential to become a standard approach in flow cytometry with many applications in biology, biophysics, biotechnology, and medicine.

## Materials and methods

### Real-time deformability cytometry

Real-time deformability cytometry (RT-DC) was carried out as described previously (*Otto et al., 2015*). For RT-DC measurements, cells were suspended in a viscosity-adjusted measurement buffer (MB) based on 1x phosphate buffered saline (PBS) containing methylcellulose. The viscosity was adjusted to 15 mPa s at room (and measurement) temperature, determined using a falling ball viscometer (Haake, Thermo Scientific). Cells in the MB were taken up into a 1 ml syringe, placed on a syringe pump (neMESYS, Cetoni GmbH) and connected via tubing to the sample inlet of the microfluidic chip with a square measurement channel cross section of $20 \times 20 \ \mu m^2$. The microfluidic chip was made from cured polydimethylsiloxane bonded to a thickness #2 cover glass. Another syringe containing MB without cells was connected to the sheath flow inlet of the chip. Measurements were carried out at a total flow rate of 0.12 µl/s with a sample flow rate of 0.03 µl/s and a sheath flow rate of 0.09 µl/s unless stated otherwise. Different gating settings for cell dimensions could be employed during the measurement (*Figure 1—figure supplement 1*). Images of the cells in the channel were acquired in a region of interest of $250 \times 80$ pixels at a frame rate of 2000 fps. Real-time analysis of the images was performed during the measurement and the parameters necessary for MORE analysis were stored for all detected cells.

### Data processing in MORE analysis

The raw data obtained from RT-DC measurements consisted of the following information of every detected cell: a bright field image of the cell, the contour of the cell, its deformation value, and the cell size as the cross-sectional area of the cell in the image (*Figure 1—figure supplement 1*). The deformation was calculated from the convex hull contour of the cell — a processed contour, where all points contributing to concave curvature were removed:

$$deformation = 1 - \frac{2\sqrt{\pi A}}{l} \ ,$$

where $A$ is the area enclosed by the convex hull contour and $l$ is the length of the convex hull contour. Therefore, deformation is the deviation from a perfectly circular cell image. It describes the change of the cell's shape by the hydrodynamic forces in the measurement channel but may also contain pre-existing shape deviations from a sphere, for example for the biconcave, disk-like shapes of healthy red blood cells or strongly activated and polarized neutrophils. Image brightness analysis was carried out using the contour information and the image of the cell. The mean brightness of the cell was determined from all pixel values within the cell's contour (*Figure 1D*). With this information the distinction of leukocyte subpopulations was possible in the space spanned by cell size and mean

brightness (*Figure 1D* and *Figure 1—figure supplement 2*). It is worth noting that the absolute value of the resulting brightness was influenced by several experimental conditions such as focus of the image and the thickness of the microfluidic chip. However, this did not affect the quality of the distinction of cells by their brightness. Special care had to be taken when comparing the brightness of different purified leukocyte subpopulations of similar size (like neutrophils, eosinophils, and monocytes). In order to achieve a situation similar to the diluted whole blood measurement, we used the same microfluidic chip repeatedly after thorough flushing. All brightness values reported were normalized to 100 by the background brightness of the channel. Apart from the initial brightness distinction, in a second step, the root mean square of pixel brightness values was calculated in an area of $9 \times 5$ pixels (nine in the flow direction, five perpendicular to the flow direction) around the geometrical center of the cell. This information was used to distinguish the relevant leukocyte subpopulations from eventual erythrocyte doublets present (*Figure 1D*). To ensure best validity of the deformation measure based on the area within the cell's contour and the length of the contour, only cells without prominent protrusions were considered for comparisons based on deformation. A reliable criterion to select those cells was found by comparing the area within the originally detected cell contour and within the convex hull contour. For erythrocytes, the difference of these two areas was limited to 15%. For leukocytes, a suitable limit was found at 5%. For the identification of malaria-infected erythrocytes we used a semi-automated procedure designed to obtain only clearly positive results and to avoid false negatives. The defining property of infected cells was the presence of bright spots within the cells. In a first step, all pixel values outside the cell's contour were set to 0. In a twice-repeated procedure, the image of the erythrocyte was further reduced by setting all pixel values of the contour pixels to 0 and finding the new contour. This measure was used to eliminate possible bright spots due to fringes at the border of the cell. From this image, the brightness of every pixel of the remaining cell was calculated by taking the mean of the pixel itself and its eight nearest neighbors. The user was then able to set the minimal threshold for this brightness in order to identify a cell as potentially infected. Since higher pixel values are frequently obtained at the rear of the cell (in flow direction) only bright spots within 70% of the cell's length counted from the front of the cell were considered. As a last criterion, the calculated brightness was compared to the brightness of the cell directly surrounding the bright spot in order to eliminate cases of generally bright cells. For this a mean brightness value was formed from all pixels located within the two rectangular areas spanned from $[k\text{-}3,l\text{-}1]$ to $[k\text{-}2,l + 1]$ as well as $[k + 2,l\text{-}1]$ to $[k + 3,l + 1]$, where $k$ is the pixel position of the bright spot in the flow direction and $l$ is the pixel position of the bright spot orthogonal to the flow direction. Most of this analysis can be performed with ShapeOut, except for the last aspect of considering details of internal brightness, for which a custom-written Python script was used.

## Blood measurements

All studies complied with the Declaration of Helsinki and involved written informed consent from all participants or their legal guardians. Ethics for experiments with human blood were approved by the ethics committee of the Technische Universität Dresden (EK89032013, EK458102015), and for human blood and LPS inhalation in healthy volunteers by the East of England, Cambridge Central ethics committee (Study No. 06/Q0108/281 and ClinicalTrialReference NCT02551614). Study participants were enrolled according to good clinical practice and recruited at the University Medical Centre Carl Gustav Carus Dresden, Germany, the Biotechnology Center, Technische Universität Dresden, Germany, or Cambridge University Hospitals, Cambridge, UK. Human blood and serum used to culture the malaria parasites was obtained from the Glasgow and West of Scotland Blood Transfusion Service; the provision of the material was approved by the Scottish National Blood Transfusion Service Committee For The Governance Of Blood And Tissue Samples For Non-Therapeutic Use. Venous blood was drawn from donors with a 20-gauge multifly needle into a sodium citrate S-monovette (Sarstedt) by vacuum aspiration. In case of blood volumes above 9 ml, blood was manually drawn via a 19-gauge multifly needle into a 40 ml syringe and transferred to 50 ml Falcon polypropylene tubes (BD) containing 4 ml 3.8% sodium citrate (Martindale Pharmaceuticals). For RT-DC measurements of blood, 50 μl of anti-coagulated blood were diluted in 950 μl MB and mixed gently by manual rotation of the sample tube. This fixed dilution of 1:20 was the result of optimization series to dilute as little possible, while still enabling the reliable detection of single cell events for both erythrocytes and leukocytes at typical cell densities found in blood. Measurements were

typically carried out within 2 hr past blood donation unless stated otherwise. Two different gating settings were employed in the measurement software for erythrocyte and leukocyte acquisition, respectively (*Figure 1—figure supplement 1A*). For erythrocytes, gates were essentially open allowing cell dimensions in flow direction from 0 μm to 30 μm. The leukocyte gate was set to a size of 5–16 μm in flow direction and >5 μm perpendicular to it. This setting allowed filtering out single erythrocytes and almost all erythrocyte multiples. The leukocyte populations remained unaltered as confirmed in experiments with purified leukocytes at open gate settings. Using the leukocyte gate, the majority of thrombocytes was also ignored as they possess typical diameters of 2–3 μm. A small fraction of very large thrombocytes and microerythrocytes were still found within this gate as seen in *Figure 1C and D*. Mechanical analysis of these events constitutes an interesting challenge in that they can be detected and counted, but at present not tested for activation via their deformation given their very small size compared to the channel size, which was chosen to accommodate all cells found in blood. Measurements in the leukocyte gate were carried out over a fixed timespan of 15 min (to acquire typically 500 to 3000 leukocytes, depending on donor and disease state), followed by a separate measurement in the erythrocyte gate for a few seconds until data of 5,000–10,000 cells were acquired. Measurements for establishing the normal MORE blood phenotype in healthy human volunteers (*Figure 1E*), and all measurements directly compared to this norm, e. g., blood samples derived from patients, were carried out at a temperature of 30°C. The remaining measurements — fMLP stimulation, LPS stimulation, purified leukocyte subpopulations, malaria infection, and erythrocyte palmitoylation — were carried out at a temperature of 23°C. The viscosity of the MB was always adjusted to 15 mPa s at the different temperatures to keep the acting hydrodynamic stress and, thus, the resulting deformation regimes the same. An MB with the viscosity of 25 mPa s (to slow blood cell sedimentation in the tubing) was used in experiments for comparing the relative cell count results of leukocyte subpopulation by MORE analysis and conventional blood count (*Figure 1F*; *Supplementary file 1*). Here, the total flow rate was 0.06 μl/s (sample flow 0.015 μl/s, sheath flow 0.045 μl/s) and images were acquired at 4000 fps.

## Leukocyte purification and identification

Leukocyte subpopulations were purified by negative and/or positive magnetic-activated cell sorting (MACS) following the instructions provided by the manufacturer. Reagents for cell isolation with magnetic beads purchased from Miltenyi Biotec were MACSxpress Neutrophil Isolation Kit human (130-104-434), Monocyte Isolation Kit human (130-091-153), Basophil Isolation Kit II human (130-092-662), Pan T Cell Isolation Kit human (130-096-535) and CD3 MicroBeads (130-050-101), as well as Pan B Cell Isolation Kit human (130-101-638) and CD19 MicroBeads (130-050-301). EasySep Human Eosinophil Enrichment Kit (19256) was obtained from StemCell Technologies. The purity of the derived cell isolates was controlled twice by staining with 7-Color-Immunophenotyping Kit (Miltenyi Biotec, 5140627058), as well as additional single staining of each cell subset for fluorescence-activated cell sorting (FACS). Individual cell type staining antibodies from BioLegend were used for granulocytes (target: CD66ACE, staining: PE, order no.: 342304, RRID:AB_2077337), eosinophils (Siglec-8, APC, 347105, RRID:AB_2561401), B lymphocytes (CD19, FITC, 302205, RRID:AB_314235), NK cells (CD56, PE, 318305, RRID:AB_604093), T helper cells (CD4, PE-Cy7, 300511, RRID:AB_314079), T lymphocytes (CD3, APC, 300411, RRID:AB_314065), cytotoxic T cells (CD8, PacificBlue, 301026, RRID:AB_493111), monocytes (CD14, FITC, 325603, RRID:AB_830676), as well as eosinophils, basophils, mast cells, and mononuclear phagocytes (CD193, PE, 310705, RRID:AB_345395). For RT-DC measurements, purified cells were pelleted by centrifugation (200 g, 5 min) and re-suspended in MB at concentrations of about $5 \cdot 10^6$ cells/ml by repeated, gentle shaking.

## In vitro malaria infection

*Plasmodium falciparum* (*P. falciparum*, HB3 clone, NCBI Taxonomy ID: 137071) cultures were grown accordingly to standard protocols (*Trager and Jensen, 1976*). Two *P. falciparum* cultures were grown independently for 3 weeks, treated with Plasmion (*Lelièvre et al., 2005*) to enrich for the schizont stages, and then allowed to reinvade fresh red blood cells in a shaking incubator for 3 hr. The cultures were then treated with sorbitol (*Lambros and Vanderberg, 1979*), to remove all schizonts that had not reached full maturity; only ring stage parasites survive sorbitol treatment. The highly synchronized culture used for the RT-DC measurements therefore consisted of erythrocytes

exposed to *P. falciparum*, into some of which parasites had invaded within a 3 hr window. Samples were removed at 4, 12, 16, 20, 24, 36, 42 and 46 hr post invasion for the RT-DC measurements. At the time of each measurement a thin blood smear was taken and stained with Giemsa's stain to assess the parasitemia and the stage of the parasites (*Figure 2—figure supplement 2A*). A control sample of the same blood without the parasites underwent the identical treatment as the *P. falciparum* exposed samples. For RT-DC measurements, at each time point, 10 µl of the blood culture were diluted in 990 µl of the MB to a final concentration of $2.5 \cdot 10^5$ cells/µl. The total flow rate through the channel was 0.04 µl/s for all malaria infection experiments (sample flow rate 0.01 µl/s, sheath flow rate 0.03 µl/s). For experiments on growth and invasion depending on erythrocyte palmitoylation status, blood, treated as described in the palmitoylation section below, was shipped from Germany to Scotland in PBS buffer containing 15 mM glucose, 5 mM sodium pyruvate, 5 µM Coenzyme A, 5 mM $MgCl_2$, 5 mM KCl, 130 mM NaCl. Parasites were synchronized by collecting *P. falciparum* mature stages (trophozoites and schizonts) from *P. falciparum* clone HB3 using MACS columns (*Ribaut et al., 2008*). The trophozoite and schizont enriched cultures were mixed with erythrocytes to achieve a starting parasitemia of 0.5–1.0%. Each erythrocyte type was set up in a separate culture flask at 3 ml volume and 5% hematocrit. The parasites were incubated in a shaking incubator at 37°C under standard culture conditions of gas and medium. Parasitemia was monitored on day 2 (post invasion) and day 4 (second round of invasion). For all experimental conditions, a minimum of 500 RBCs were counted. Experiments were repeated on three different days with erythrocytes of 3 different donors yielding the same results.

## Palmitoylation of erythrocytes

Red blood cells were pelleted by blood centrifugation (800 g, 5 min), plasma was removed, and the RBCs were pretreated with one volume of 1% fatty acid-free bovine serum albumin (BSA) in PBS-glucose (10 mM phosphate, 140 mM NaCl, 5 mM KCl, 0.5 mM EDTA, 5 mM glucose, pH 7.4) at 37°C for 15 min, in order to lower the endogenous content of free fatty acids in their membrane pools, and washed three times with PBS-glucose. Cells were re-suspended in 3 volumes of incubation buffer, containing 40 mM imidazole, 90 mM NaCl, 5 mM KCl, 5 mM $MgCl_2$, 15 mM D-glucose, 0.5 mM EGTA, 30 mM sucrose, 5 mM sodium pyruvate, 5 mM Coenzyme A, 50 mg PMSF/ml and 200 U penicillin/streptomycin (320 mOsm, pH 7.6). For inhibition of palmitoylation, 100 µM final concentration of 2-bromopalmitate (2 BP) was used. 100 µM palmitic acid (PA) was added as a control. The RBCs were incubated in a humidified incubator with 5% $CO_2$ for 24 hr at 37°C. Prior to measurement, RBCs were pelleted, re-suspended in 1% BSA, incubated for 15 min at 37°C and washed two times with PBS-glucose. Glucose, sucrose, 2-bromopalmitate, palmitic acid, fatty acid free BSA, Coenzyme A, and PMSF were purchased from Sigma-Aldrich; Penicillin/streptomycin and sodium pyruvate from Gibco. RT-DC measurements were carried out at a room temperature of 23°C and with a total flow rate of 0.032 µl/s (sample flow 0.008 µl/s, sheath flow 0.024 µl/s) after adding 10 µl of the RBC suspension to 990 µl of MB. Experiments were carried out on two different days with erythrocytes of 4 different donors.

## fMLP-induced neutrophil activation

For in vitro fMLP stimulation, blood was stimulated with 100 nM N-Formylmethionyl-leucyl-phenylalanine (fMLP; Sigma-Aldrich, 47729, 10 mg-F). Separate samples were analyzed in time intervals of 0–15 min, 15–30 min, 30–45 min, and 45–60 min after activation. During incubation all samples were stored in 2 ml Eppendorf tubes at 37°C at 450 rpm in a ThermoMicer C (Eppendorf). All experiments were performed within 2 hr maximum after blood drawing. Experiments were repeated with blood samples of 5 different donors on five different days. Due to experimental feasibility PBS controls of these donors were measured before fMLP stimulation and after the 60 min fMLP sample. Additionally, three control samples of different donors were treated similarly adding 10 µl 1 x PBS instead of fMLP and were analyzed in time intervals of 0–15 min, 15–30 min, 30–45 min, and 45–60 min after bleeding to exclude kinetic effects due to blood alteration with storage.

## In vitro *Staphylococcus aureus* infection

Blood stimulation was performed with *Staphylococcus aureus* Newmann strain (*S. aureus*; ATCC 25904; NCBI Taxonomy ID: 426430). For reproducible repetitive testing with competent bacterial

strains cryo-aliquots of *S. aureus* were prepared as follows. Bacterial cells were pre-cultured to the log phase for synchronization of growth in BHI broth (Bacto Brain Heart Infusion, Becton Dickinson) at 37°C and transferred to a second culture. Aiming at a high bacterial virulence factor expression, the cells were grown to an early stationary phase in a 96-well-plate (100 µl, $OD_{600nm}$ 0.1837, Infinite 200 reader, TECAN), pelleted by centrifugation (2671 g for 5 min at 4°C), washed two times in PBS and re-suspended in cell-freezing media (Iscove Basal Medium, Biochrom) with 40% endotoxin-free FBS (FBS Superior, Biochrom) at a final concentration of $2.54 \cdot 10^9$ CFU/ml. Aliquots were immediately frozen at –80°C and only thawed once for a single experiment. Blood stimulation and measurement were carried out at 30°C temperature for 15 min with one multiplicity of infection (MOI) in 1:20 RT-DC measurement buffer. MOI (0.9–1.09) was controlled retrospectively by granulocyte count and 5% sheep blood agar culture (Columbia agar, bioMérieux) at 37°C and bacterial colony counting on the following day. PBS blood controls were conducted before and after *S. aureus* blood stimulation. The experiment was repeated with blood of 4 different donors on four different days. All experiments were performed within 2 hr after blood drawing.

## LPS inhalation

*E. coli* lipopolysaccharide (LPS) 50 µg (GSK) was administered to healthy, never-smoker volunteers via a specialized dosimeter (MB3 Markos Mefar) 90 min prior to injection of autologous [99m]Technetium-Hexamethylpropleneamine-oxime labeled neutrophils. Temperature, forced expiratory volume in 1 s, forced ventilator capacity and triplicate blood pressures were recorded prior to, and at 30 min post LPS administration. RT-DC measurements were obtained at baseline, 90, 135, 210, 330, and 450 min post LPS.

## Respiratory tract infections (RTI) and acute lung injury (ALI)

Patient inclusion criteria for RTI: Patients with clinical signs of lower RTI, a core temperature >38.5°C and the need for supplemental oxygen were recruited on the day of hospitalization. Only patients without treatment prior to hospitalization were included. None of the included patients received antibiotic treatment for reconstitution. Patient inclusion criteria for ALI: Patients diagnosed with ALI according to the criteria of the North American European Consensus Conference (NAECC) (*Bernard et al., 1994*) and without underlying diseases prior to ALI were included. All blood samples were analyzed within 30 min of venipuncture. Size and deformation of blood leukocytes was characterized for all blood cells in which the area within the original cell contour differed less than 5% from the area within the convex hull contour.

## Acute myeloid/lymphatic leukemias

Samples from patients diagnosed with ALL or AML based on cytogenetic, molecular-genetic and morphological criteria according to WHO classification from 2008 (*Vardiman et al., 2009*) were assessed by MORE blood analysis on the day of diagnosis. In order to evaluate mechanical properties of AML and ALL blast cells in diluted whole blood, several brightness and size gates had to be combined as shown in *Figure 4A–C*. The AML gate spanned the regions normally used for basophils and monocytes. The ALL gate spanned the regions used for lymphocytes, basophils and monocytes. In all AML cases, blasts made up >80% of all leukocytes, and up to 99% of events in the AML gate. In all ALL cases, blasts made up >60% of leukocytes, and up to 85% of events in the ALL gate. The blast cell fraction was obtained from the standard differential blood count, by comparing the number of blast cells with the number of normal cells that would also populate the respective blasts gate in MORE analysis.

## Isoelasticity lines and young's moduli

RT-DC data of cell size and deformation can be converted into apparent Young's moduli using theoretical models (*Mietke et al., 2015*) and numerical simulations (*Mokbel et al., 2017*). To ensure a correct conversion, effects of shear thinning of the MC medium and a deformation offset due to pixelation were taken into account as described in *Herold, 2017*. The calculation of apparent Young's moduli for AML and ALL blasts and isoelasticity lines are based on the assumption that cells can be approximated as purely elastic, homogeneous isotropic spheres. This assumption is equivalent to using the Hertz model to extract an apparent Young's modulus of cells in atomic force microscopy-

enabled nano-indentation experiments. The conversion of deformation and size into Young's modulus for every cell measured is included in the analysis software ShapeOut.

## Statistics

Throughout, the number of cells in a single measurement is denoted as $N$, while the number of independently repeated experiments — typically the number of donor or patient samples measured, as stated — is denoted as $n$. For comparison of different donors or treatment conditions the median of deformation and cell size of a specific cell population was used. In order to evaluate effects of a disease we calculated a 2D confidence ellipse at 68.3% (or one sigma) as well as 95.5% (or two sigma) for the control group/norm norm of healthy human blood donors in the space of cell size and deformation. The confidence ellipse was calculated from the covariance matrix of the data and the calculation was carried out with OriginPro 2015 (Originlab). Statistically significant differences between two sets of experiments were checked to the significance level of $p<0.05$ by comparing the groups of individual median values of an experiment using a Kruskal-Wallis one-way ANOVA as implemented in OriginPro 2015 (Originlab). In erythrocyte MORE analysis in malaria infection and palmitoylation, statistically significant differences were checked using linear mixed models in combination with a likelihood ratio test to obtain significance levels for the comparison of the complete populations (*Herbig et al., 2017*). This analysis can be performed in the software ShapeOut. One, two, or three asterisks were awarded for significance levels $p<0.05$, $p<0.01$ and $p<0.001$, respectively. In manual counts of malaria infection in RBCs, statistical analyses were performed using a $\chi^2$ test with Bonferroni correction (adjusted statistical significance for $p<0.0125$) to compare the numbers of infected and non-infected erythrocytes between erythrocyte samples, except where number of parasite infected cells was zero, in which case Fisher's exact test was used. The standard deviation for the parasitemia was calculated assuming a binomial random variable as $SD = \sqrt{N \cdot p\,(1-p)}$, where $N$ is the number of cells counted and $p$ is the fraction of infected cells.

## Data availability

The raw data of all measurements are available from the Dryad Digital Repository (*Toepfner et al., 2017*). The TDMS files can be read, processed, and analyzed using ShapeOut, a custom written, open source software.

## Code availability

RT-DC measurement software is commercially available. The analysis software ShapeOut is available as an open source application on GitHub (https://github.com/ZELLMECHANIK-DRESDEN/ShapeOut/releases; *Müller, 2017*). A copy is archived at https://github.com/elifesciences-publications/ShapeOut.

## Acknowledgements

The authors would like to thank Björn Lange, Michael Mögel, Beate Eger, Isabell Deinert, Tamara Schön and the whole team for their contributions to patient recruitment, Claudia Bratsch for cell isolation, Uta Falke and Isabel Richter for technical help, Thomas Krüger for help with drawing blood, Ramona Hecker for technical advice, Mike Blatt for the loan of a microscope, Elizabeth Peat for assistance with malaria parasite culture materials, Salvatore Girardo of the BIOTEC/CRTD Microstructure Facility (in part funded by the European Fund for Regional Development – EFRE) for help with preparation of PDMS chips, and Stephan Grill for critical reading of the manuscript.

## Additional information

### Competing interests

Christoph Herold: Owns shares of, and is full-time employed at, Zellmechanik Dresden GmbH, a company selling devices based on real-time deformability cytometry. The author has no other financial interests to declare. Oliver Otto, Philipp Rosendahl: Own shares of, and are part-time employed at, Zellmechanik Dresden GmbH, a company selling devices based on real-time deformability

cytometry. The author has no other financial interests to declare. Zellmechanik Dresden GmbH did not have any role in the conception and planning of this study, or its preparation for publication. The other authors declare that no competing interests exist.

## Funding

| Funder | Grant reference number | Author |
| --- | --- | --- |
| Alexander von Humboldt-Stiftung | Alexander von Humboldt Professorship | Jochen Guck |
| Deutsche Forschungsgemeinschaft | TRR83 and SFB655 | Ünal Coskun Martin Bornhäuser |
| Seventh Framework Programme | ITN | Lisa Ranford-Cartwright Birgitta Henriques-Normark Jochen Guck |
| Bundesministerium für Bildung und Forschung | German Center for Diabetes Research (DZD e.V.) | Ünal Coskun |
| Sächsisches Staatsministerium für Wissenschaft und Kunst | TG70 AZ 4-7531.60/29/45 | Oliver Otto Jochen Guck |
| Tour der Hoffnung | Non-commercial grant | Julia Stächele |
| Sonnenstrahl e.V. Dresden | Non-commercial grant | Meinolf Suttorp |
| Zentrum für Regenerative Therapien Dresden | Seed grant FZ 111 | Jochen Guck |
| Technische Universität Dresden | Support the Best Program | Reinhard Berner Jochen Guck |
| National Institute for Health Research | Cambridge Biomedical Research Centre | Edwin R Chilvers |
| GlaxoSmithKline | Non-commercial grant | Edwin R Chilvers |
| Seventh Framework Programme | ERC Starting Grant #282060 | Jochen Guck |

The funders had no role in study design, data collection and interpretation, or the decision to submit the work for publication.

## Author contributions

Nicole Toepfner, Conceptualization, Data curation, Formal analysis, Validation, Investigation, Methodology, Writing—original draft, Writing—review and editing; Christoph Herold, Conceptualization, Data curation, Software, Formal analysis, Validation, Investigation, Visualization, Methodology, Writing—original draft, Writing—review and editing; Oliver Otto, Software, Formal analysis, Supervision, Funding acquisition, Methodology; Philipp Rosendahl, Data curation, Software, Formal analysis, Investigation, Methodology; Angela Jacobi, Assisted with cell isolation experiments and performed the leukemia experiments; Martin Kräter, Laura Ciuffreda, Resources, Validation, Investigation; Julia Stächele, Funding acquisition, Investigation; Leonhard Menschner, Elisabeth Reithuber, Geneviève Garriss, Investigation; Maik Herbig, Programmed analysis methods, Conducted malaria experiments; Lisa Ranford-Cartwright, Ünal Coskun, Conceptualization, Resources, Supervision, Funding acquisition, Writing—review and editing; Michal Grzybek, Conceptualization, Resources, Investigation, Writing—review and editing; Peter Mellroth, Supervision, Investigation; Birgitta Henriques-Normark, Conceptualization, Supervision, Funding acquisition, Writing—review and editing; Nicola Tregay, Resources, Investigation; Meinolf Suttorp, Resources, Supervision, Funding acquisition, Writing—review and editing; Martin Bornhäuser, Conceptualization, Resources, Supervision, Writing—review and editing; Edwin R Chilvers, Reinhard Berner, Conceptualization, Resources, Supervision, Funding acquisition, Project administration, Writing—review and editing; Jochen Guck, Conceptualization, Supervision, Funding acquisition, Methodology, Writing—original draft, Project administration, Writing—review and editing

## Author ORCIDs

Philipp Rosendahl (iD) http://orcid.org/0000-0002-9545-5045

Martin Kräter (iD) http://orcid.org/0000-0001-7122-7331

Ünal Coskun (iD) http://orcid.org/0000-0003-4375-3144

Geneviève Garriss (iD) https://orcid.org/0000-0002-5361-0975

Edwin R Chilvers (iD) http://orcid.org/0000-0002-4230-9677

Jochen Guck (iD) http://orcid.org/0000-0002-1453-6119

## Ethics

Human subjects: The work involved measurements of human blood samples. All studies complied with the Declaration of Helsinki and involved written informed consent from all participants or their legal guardians. Ethics for experiments with human blood were approved by the ethics committee of the Technische Universität Dresden (EK89032013, EK458102015), and for human blood and LPS inhalation in healthy volunteers by the East of England, Cambridge Central ethics committee (Study No. 06/Q0108/281 and ClinicalTrialReference NCT02551614).

## Decision letter and Author response

Decision letter https://doi.org/10.7554/eLife.29213.031

Author response https://doi.org/10.7554/eLife.29213.032

---

## Additional files

### Supplementary files

• Supplementary file 1. Table of relative blood counts by MORE analysis and conventional analysis. Percentage of all leukocytes identified by MORE analysis compare to conventional full blood cell counts, obtained with Sysmex XE-5000 differential analyzer and verified by a microscopic differential count, of four donors, two male (A, C), two female (B, D). The absolute cell counts per volume obtained by MORE analysis differ from the values of the conventional blood count, since some cells are not detected (up to 40% of all cells). However, this affects all leukocytes similarly so that the relative counts are not changed.

DOI: https://doi.org/10.7554/eLife.29213.025

• Supplementary file 2. Table comparing conventional biomarkers of leukemia with MORE analysis. (1) Morphological analysis of air-dried Romanowsky (Wright, Wright-Giemsa, or May-Grünwald-Giemsa)-stained blood or bone marrow smears. The morphological features identified by microscopic examination may suggest either lymphoid or myeloid differentiation of leukemic cells, but with the exception of the identification of Auer rods in myeloblasts none of these features is lineage-specific. Sub-clones can be identified by differences in size and morphological features (e. g. cytoplasmatic vacuoles). (2) Cytochemical staining improves the accuracy and reproducibility of lineage assessment and therefore is required for traditional sub-classification of acute myeloid leukemia (AML) according to the French-American-British (FAB) and WHO criteria. Sudan Black and stains for myeloperoxidase (MPO) to identify myeloblasts and esterase stains like alpha-naphthyl-butyrate to identify monoblasts have remained useful in this regard. Staining must be performed without undue delay as MPO is unstable and becomes undetectable after a week of storage. (3) Immunophenotypic classification is based on identification of cell surface epitopes or cytoplasmatic proteins by fluorescent dye-labeled antibodies. Flow cytometry (fluorescence-activated cell sorting, FACS) is nowadays widely used as a particularly powerful method because multiparameter analysis offers the advantage of segregating leukemic cells from non-neoplastic cells. Thus, rapid analysis allows to establish the lineage of the leukemia (e.g. myeloid versus lymphoid), its stage of differentiation (e. g. T- versus B-ALL) and facilitates minimal residual disease (MRD) monitoring using a leukemia-specific pattern of markers not expressed in that combination on regular blood or bone marrow cells. Notably, some precursor B-cell ALL might be negative for CD45 (leukocyte common antigen) or patients with T-ALL lack TdT or CD34 expression. Although ALL can be classified according to the stage of maturation, the optimal immunologic sub-classification remains a matter of debate. Many ALLs also aberrantly express myeloid-linage associated antigens (mostly CD13, CD33). Therefore the antibody screening panel for acute leukemias must be designed to include at least one very sensitive and one relatively

specific marker for each hematopoietic and lymphoid lineage. (4) Molecular (genetic) classification using traditional methods will detect specific cytogenetic and/or molecular abnormalities in 60–80% of ALL and 50–60% of AML cases. The recent advent of whole genome analysis has allowed virtually all acute leukemias to be classified according to specific genetic abnormalities. Markers can be separated into leukemia-specific (e.g. BCR-ABL1; t(15;18)) or leukemic-clone specific (e.g. Ig-heavy chain gene rearrangements, T-cell receptor gene rearrangements). Both are valuable for classification, as prognostic indicators with a defined treatment applied, and are nowadays routinely used for monitoring of MRD by exploiting the high sensitivity of PCR-based amplification of specific gene sequences. The technique is time-consuming and expensive, and usually performed only in reference laboratories. (5) MORE analysis. When compared to these established conventional methods, the advantages of morpho-rheological (MORE) phenotyping are characterized by a very short time for analysis and the minimum amount of blood required. The technique has comparable power with regard to the identification of leukemic cells and the identification of leukemic sub-clones. Its applicability to classify the leukemic lineage (for example by significant differences in size, deformation, and Young's modulus; see *Figure 4—figure supplement 1*) and to detect small numbers of leukemic cells can theoretically be expected and has been shown in single cases already, but still has to be tested and proven in a formal comparison, which is beyond the scope of the present study. Potentially, the rheological features of blast cells might represent additional prognostic biomarkers for leukemic cells (stiffness might correlate to drug sensitivity or refractoriness, or identify a leukemic subclone), which will be the subject of future studies. Morpho-rheological phenotyping, thus, compares very well to established biomarkers for following ALL treatment success.

DOI: https://doi.org/10.7554/eLife.29213.026

• Transparent reporting form

DOI: https://doi.org/10.7554/eLife.29213.027

## Major datasets

The following dataset was generated:

| Author(s) | Year | Dataset title | Dataset URL | Database, license, and accessibility information |
|---|---|---|---|---|
| Toepfner N, Herold C, Otto O, Rosendahl P, Jacobi A, Kräter M, Stächele J, Menschner L, Herbig M, Ciuffreda L, Ranford-Cartwright L, Grzybek M, Coskun Ü, Reithuber E, Garriss G, Mellroth P, Normark BH, Tregay N, Suttorp M, Bornhäuser M, Chilvers E, Berner R, Guck J | 2017 | Data from: Detection of human disease conditions by single-cell morpho-rheological phenotyping of whole blood | https://doi.org/10.5061/dryad.2fk71 | Available at Dryad Digital Repository under a CC0 Public Domain Dedication |

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
