## [Decision Letter]

Thank you for submitting your article "Detection of human disease conditions by single-cell morpho-rheological phenotyping of whole blood" for consideration by *eLife*. Your article has been reviewed by three peer reviewers, and the evaluation has been overseen by a Reviewing Editor and Ivan Dikic as the Senior Editor. The following individuals involved in review of your submission have agreed to reveal their identity: Dennis Discher (Reviewer #1); Amy C Rowat (Reviewer #3).

The reviewers have discussed the reviews with one another and the Reviewing Editor has drafted this decision to help you prepare a revised submission. The editors and reviewers agree that this work is interesting and potentially quite useful however, as your effort is directed at the development of a new technique rather than a discovery with novel conclusions, we feel it will be more appropriately published as a Tools and Resources paper rather than as a Research Article.

Summary:

Blood cell numbers, sizes, and deformability have been quantified for many decades at the single cell level under normal and numerous diseased conditions for 2-3 decades or more, although throughput for deformation measures is more limited. This manuscript describes a microfluidic approach to the above quantitation, called MORE (cell-by-cell morpho-rheological) analysis, capable of treating around 1000 blood cells/s, from 10 µL drop of blood only, adding deformation information for each cell type. The principle of the method is based on image acquisition by high-speed video-microscopy and post-acquisition image analysis. The method correctly phenotypes different blood cells using both their typical projected area in the constriction as well as their compactness or their deformation. Adding a third parameter for classification represented by the "brightness" of the cells, the authors are successfully able to reproduce most of the classical and simple hematological phenotyping present on the market of automated blood counts without any staining steps. Moreover, the authors provide a comprehensive set of examples from leukemia patients, or malaria infection, bacterial and viral infections as well as donor samples with anti-coagulants.

Overall, this manuscript is well written, interesting and timely. Its strength is clearly the technology, with an impressive collection of data validating expectations, and only a few key concerns temper enthusiasm. We have the following suggestions to improve the manuscript.

Essential revisions:

1) The authors over-use the acronym MORE, in particular in the captions. They should replace these with text that highlights the new biological finding or concept that is learned. Moreover, the authors should highlight what is scientifically new for the well-established field of hemorheology.

2) Statistical significance and confidence should be clarified at different places.

- Figure 3 the number of EBV cells is not provided. Also, authors define n as number of independent experiments, does this reflect the number of patients or the same patient cells measured over independent experiments? It would be helpful to show single cell distributions for EBV given the focus of the paper.

- The authors need to comment on the utility and statistical significance of deformation measurements in Figure 4 for the 4-6 patients with AML or ALL, since the deformation data adds little to nothing beyond cell size measurements.

- Given that Figure 4 is a time series generated from just one patient, it is at least important to provide measures of statistical significance and confidence. Each datapoint is the median for a given day, and so error bars should be added for each day. An asterisk could be added to each datapoint that differs from the first, or some other scheme. More important, given that this is a new method, some type of daily standard (i.e. normal control) should have been run and shown in parallel with the patient measurements. Knowledge of how this data compare to conventional biomarkers/existing methods for analysis following treatment would be helpful to benchmark RTDC.

3) Figure 2 shows control vs exposed but then data shown in Figure 2 also includes infected. The difference and disease relevance of exposed vs infected populations should be clarified for the reader.

The authors claim the greater deformation reduction of infected cells vs entire exposed population (Figure 2), which may be explained by clearance of stiff cells by the spleen. But if stiffer cells are filtered out by the spleen, this would result in a lower deformation reduction.

It is unclear why the 2BP and PA treatments were performed. It would be helpful for the reader to clarify the motivation for these treatments as they relate to pf infection.

4) Subsection “MORE analysis of leukocytes” what is the timepoint of mechanical measurements of activated neutrophils published in the older reports? If measurements were performed at different time points, with different methods that deform cells on vastly different timescales, it does not seem to be a conflict.

Neutrophils were less deformable after fMLP treatment, but became more deformable and larger at longer time points. Given that 'larger cells of identical stiffness should deform more in RTDC' (subsection “MORE analysis of malignant transformed blood cells”), how can these results show that fMLP cells, which are bigger, are less deformable? If larger cells are deforming more in RTDC (and when deformed by inertial flows as in Gossett et al.), then it is difficult to compare to previous methods.

---

## [Author Response]

1) The authors over-use the acronym MORE, in particular in the captions. They should replace these with text that highlights the new biological finding or concept that is learned. Moreover, the authors should highlight what is scientifically new for the well-established field of hemorheology.

We have to admit that we quite like the acronym MORE and might have gotten carried away, but of course see the point that it might appear overused to the unbiased reader. We have removed it from all captions, and moderated its use throughout.

The contribution of our method to the well-established field of hemorheology is that it can provide mechanical data of all individual blood cells directly in whole blood and without prior separation. This permits a much more specific investigation of the individual contribution of each blood cell type to the overall rheological properties of blood, which was not possible so far. We have further emphasized this aspect in the revised manuscript.

2) Statistical significance and confidence should be clarified at different places.- Figure 3 – The number of EBV cells is not provided. Also, authors define n as number of independent experiments, does this reflect the number of patients or the same patient cells measured over independent experiments? It would be helpful to show single cell distributions for EBV given the focus of the paper.

There might have been a mix-up in the figure panel mentioned. Figure 3 actually shows a measurement example of the population of neutrophil granulocytes of a healthy donor and a patient suffering from respiratory tract infection (not EBV). For both, the number of individual cells measured is stated directly in the legend.

Generally, “n” is the number of independent measurement repeats. If this specifically means different donors/patients, we now explicitly state this at all relevant places in the manuscript.

We fully agree that single cell distributions of a typical EBV patient blood measurement were missing, and showing them improves the paper by being maximally transparent. We therefore added exemplary single cell distributions for neutrophils, lymphocytes, and monocytes of a healthy donor and patients with ALI, RTI, and EBV as new Figure 3—figure supplement 4. Please note that in addition to the exemplary distributions shown in the manuscript, all raw data – underlying the further processed plots, which only contain the medians for clarity – are available as TDMS files upon request, and can be analyzed by the open source software ShapeOut. This has now been stated explicitly in the methods section.

- The authors need to comment on the utility and statistical significance of deformation measurements in Figure 4 for the 4-6 patients with AML or ALL, since the deformation data adds little to nothing beyond cell size measurements.

The diagnostic utility of the morpho-rheological analysis of leukemia patient blood is foremost the identification of unusual cell populations (immature blast cells) in whole blood (see Figure 4), which will be useful to aid and supplement initial diagnosis. Once identified by appropriate gating, it is then possible to also consider their mechanical properties (deformation vs. size plots, Figure 4). This is at present primarily aimed at establishing a connection to the established field of cancer cell mechanics, which so far mostly deals with solid tumors. The few experiments done on leukemic cells were only possible either for immortal cell lines or for specifically isolated cells (see relevant references in text). Our results confirm the composite picture emerging from these previous studies — that both lymphatic and myeloid blast cells are stiffer (!) than their healthy counterparts — for the first time in real patient samples and without prior isolation. We realize that the description of these results might have been unclear and confusing, referring to how much cells should have deformed or not at the same or different sizes. Enabled by recent progress in the numerical modeling of cell deformation in narrow channels (see Mokbel et al., 2017), we can now also calculate the apparent Young’s modulus of leukocytes. This was not possible before with the analytical model (Mietke et al., 2015) as leukocytes were too small and/or too deformed to be captured by the model. We have now added isolelasticity lines between 0.55 kPa and 1.1 kPa to Figure 4 for better parameterization of the deformation-size space and in order to stress the mechanical difference between ALL and AML blast cells compared to the leukocyte subtypes of the norm, as well as compared to each other. In addition to the existing panel of Figure 4—figure supplement 1, which compares deformation and cell size of ALL blasts with lymphocytes as well as AML blasts with neutrophils, monocytes, and eosinophils, a new panel was added comparing the apparent Young’s moduli of these cells, including their statistical significance. The methods section was extended to describe the calculation of Young’s moduli and the main text was changed accordingly to clarify the statement about the mechanical differences. We hope that this addresses the reviewers’ point, but want to stress that the main focus of this manuscript is not the detailed mechanical characterization of individual cells (which we now have added for specific examples upon the reviewers’ request) but the identification of patterns visible in the deformation-size scatter plots — the morpho-rheological fingerprint — to identify disease conditions.

- Given that Figure 4 is a time series generated from just one patient, it is at least important to provide measures of statistical significance and confidence. Each datapoint is the median for a given day, and so error bars should be added for each day. An asterisk could be added to each datapoint that differs from the first, or some other scheme. More important, given that this is a new method, some type of daily standard (i.e. normal control) should have been run and shown in parallel with the patient measurements. Knowledge of how this data compare to conventional biomarkers/existing methods for analysis following treatment would be helpful to benchmark RTDC.

We agree that a measure of confidence for each individual data point shown in Figure 4 would be useful and appropriate. However, simply adding error bars to the median points or calculating a statistical significance of the difference between two data points is unsuitable for the following reasons:

i) As one data point of Figure 4 is the median of a population of 1,000 to 13,000 cells measured, this value is highly precise. Error bars displaying the standard error of the mean (or equivalent) would disappear behind the median symbols as the relative error is always less than 1% (and in most cases much less than 1%) in both directions – cell size and deformation. Furthermore, displaying the error bars as standard deviation would not add information about either confidence of statistical significance. This would only add information about the spread of the individual populations, information that can also be found in the actual distributions shown in panels E to G, and it would make panel H difficult to read.

ii) For the same reason of very well defined populations with thousands of underlying data points, any difference between two measurements will always be highly significant. As an example, the measurements of day 7 and day 8 (which are the two points closest to each other in this panel) are statistically significantly different both in cell size and deformation with a p value < 10^–4^. This shows very clearly the unprecedented power of the technique, say compared to AFM measurements. The sheer number of measurements taken makes any difference significant.

Instead, and as also suggested by the reviewers, a normal standard for the reliability of a sample measurement would be more important and relevant. While we do not have measurements of normal control samples taken on the same days as the patient samples, we did perform extensive control measurements for the stability and reliability of the method (see Figure 1 and Figure 1—figure supplement 4,Figure 1—figure supplement 5,Figure 1—figure supplement 6). These can be used to obtain a confidence region for a single measurement. This confidence region takes into account the behavior of a certain cell type and the variation between repeated measurements of the same donor on a technical level (such as sample handling) and on the biological level as well. Such confidence regions can be calculated from the repeated measurements of the two donors displayed in Figure 1—figure supplement 6. Panels G-I were added to this figure which show the 68% and 95% confidence regions of a single experiment for neutrophils, monocytes, and lymphocytes. The confidence region for lymphocytes was then used and added to Figure 4 at four data points (day 1, 4, 8, and 12) in order to allow for a better judgment of the degree of the change during the evolution of this time course, and at the same time to preserve the legibility of the figure.

Conventional biomarkers and existing methods for the analysis of leukemia are the following:

1) Morphological analysis – of air-dried Romanowsky (Wright, Wright-Giemsa, or May-Grünwald- Giemsa)-stained blood or bone marrow smears. The morphological features identified by microscopic examination may suggest either lymphoid or myeloid differentiation of leukemic cells, but with the exception of the identification of Auer rods in myeloblasts none of these features is lineage-specific. Subclones can be identified by differences in size and morphological features (e. g. cytoplasmatic vacuoles). The time required for analysis comprises 1 – 2 hrs. Amount of material required: 100 µl. The costs are relatively low (estimate: 5 €).

2) Cytochemical staining – improves the accuracy and reproducibility of lineage assessment and therefore is required for traditional sub-classification of acute myeloid leukemia (AML) according to the French-American-British (FAB) and WHO criteria. Sudan Black and stains for myeloperoxidase (MPO) to identify myeloblasts and esterase stains like α-naphthyl-butyrate to identify monoblasts have remained useful in this regard. The time required for analysis comprises 1 – 2 hrs. Staining must be performed without undue delay as MPO is unstable and becomes undetectable after a week of storage. Amount of material required: 100 µl. The costs are relatively low (estimate: € 5 – 10).

3) Immunophenotypic classification – is based on identification of cell surface epitopes or cytoplasmatic proteins by fluorescent dye-labeled antibodies. Flow cytometry (FACS analysis) is nowadays widely used as a particularly powerful method because multiparameter analysis offers the advantage of segregating leukemic cells from non-neoplastic cells. Thus, rapid analysis allows to establish the lineage of the leukemia (e.g. myeloid versus lymphoid), its stage of differentiation (e. g. T- versus B-ALL) and facilitates minimal residual disease (MRD) monitoring using a leukemia-specific pattern of markers not expressed in that combination on regular blood or bone marrow cells. Notably, some precursor B-cell ALL might be negative for CD45 (leukocyte common antigen) or patients with T-ALL lack TdT or CD34 expression. Although ALL can be classified according to the stage of maturation, the optimal immunologic sub-classification remains a matter of debate. Many ALLs also aberrantly express myeloid-linage associated antigens (mostly CD13, CD33). Therefore, the antibody screening panel for acute leukemias must be designed to include at least one very sensitive and one relatively specific marker for each hematopoietic and lymphoid lineage. Amount of material required: 100 µl – 1ml. The method can require 1 – 4 hrs of time. The costs have to be considered depending on the quantity of antibodies used (estimate: € 100 – 500).

4) Molecular (genetic) classification – using traditional methods will detect specific cytogenetic and/or molecular abnormalities in 60 – 80% of ALL and 50 – 60% of AML cases. The recent advent of whole genome analysis has allowed virtually all acute leukemias to be classified according to specific genetic abnormalities. Markers can be separated into leukemia-specific (e.g. BCR-ABL1; t(15;18)) or leukemic-clone specific (e.g. Ig-heavy chain gene rearrangements, T-cell receptor gene rearrangements). Both are valuable for classification, as prognostic indicators with a defined treatment applied, and are nowadays routinely used for monitoring of MRD by exploiting the high sensitivity of PCR-based amplification of specific gene sequences. The technique is time-consuming (1 – 10 days) and expensive (estimate: € 500 – 5,000) and usually performed only in reference laboratories.

As outlined in table 1 below, when compared to these established conventional methods the advantages of morpho-rheological phenotyping are represented by a very short time for analysis and the minimum amount of blood required. The technique has comparable power with regard to the identification of leukemic cells and the identification of leukemic sub-clones. Its applicability to classify the leukemic lineage (for example by significant differences in size, deformation, and Young’s modulus; see Figure 4—figure supplement 1) and to detect small numbers of leukemic cells can theoretically be expected and has been shown in single cases already, but still has to be tested and proven in a formal comparison, which is beyond the scope of the present study.

Table 1: Comparison of conventional markers of leukemia with MORE analysis

**Issue of**

**analysis**
**Metho-**
**dology**Identifi-cation of leukemic cellsIdentification of leukemic subclonesClassifi-cation of leukemic lineageLower limit for detection of MRDAmount of material requiredTime required for ana-lysisEst. cost of ana-lysis [€]**Morpho-logical analysis**YesOnly if based on morpho-logical differ-rencesUncertain1 / 1x10^2^100 µl1 – 2 h5**Cyto-chemical**
**staining**YesYesYes1 / 1x10^2^100 µl1 –2 h10**FACS analysis**YesYesYes1 / 5x10^3^
– 1x10^4^100 – 1000 µl1 – 4 h100 – 500**Molecular analysis**YesYesYes1/ 1x10^4^
– 1x10^6^5 – 10 ml1 – 10 days500 – 5000**MORE analysis**YesYesPossible, but not yet formally testedPossible, but not yet formally tested10 µl30 min20

Abbreviations: FACS Fluorescence activated cell scanning; MRD minimal residual disease; MORE morpho-rheological

Response to treatment is one of the most powerful prognostic “in vivo” markers of leukemia survival. In pediatric ALL the number of blasts at day 8 after start of methylprednisolone treatment is predictive of the relapse rate (< 1,000 blasts/µl of blood: relapse rate 20 – 30%; >1,000 blasts/µl of blood: relapse rate 50 – 80%). As only the morphological analysis is used to monitor the effects of methylprednisolone treatment in ALL over the first 12 days at present, MORE analysis provides at least the same information about the treatment success in a shorter amount of time and with smaller sample sizes required. Potentially, the rheological features of blast cells might represent additional prognostic biomarkers for leukemic cells (stiffness might correlate to drug sensitivity or refractoriness, or identify a leukemic subclone), which will be the subject of future studies. Morpho-rheological phenotyping, thus, compares very well to established biomarkers for following ALL treatment success. We have added a short statement summarizing these aspects in the revised manuscript and appended the comparison table as Supplementary file 2.

3) Figure 2 shows control vs exposed but then data shown in Figure 2 also includes infected. The difference and disease relevance of exposed vs infected populations should be clarified for the reader.

We thank the reviewers for this comment, which helps us clarifying the experimental procedure. What we call “exposed” is the RBC sample with vital *Plasmodium falciparum* parasites present in the sample. Of all the RBCs, only a small percentage of cells is actually infected by a parasite (the percentage of infected cells is called parasitemia and given in percent). This is true for both our in vitro experiment, as well as the disease-relevant situation in vivo (even though parasitemias in vivo are usually smaller than used here). So, Figure 2 actually shows the distribution of deformation and size in an exposed population (exposed cells shown as blue dots), with the few infected cells (identified by the white image feature shown in the inset) labeled by the red dots. Figure 2 then shows the analysis of median deformation of all control (black, Figure 2), exposed (blue, Figure 2), and actually infected (red, Figure 2) cells over the 46h course of the experiment. We have relabeled the x-axis of Figure 2 (and of Figure 2—figure supplement 2) as “incubation time” (rather than “time post infection”) and modified the main text, the figure caption, and the methods description to make the difference between exposed and infected clearer.

The authors claim the greater deformation reduction of infected cells vs entire exposed population (Figure 2), which may be explained by clearance of stiff cells by the spleen. But if stiffer cells are filtered out by the spleen, this would result in a lower deformation reduction.

This is a valuable remark. The apparent contradiction can be resolved by pointing out that our experiments were done in vitro and we did not analyze blood actually taken from malaria patients. The comment about removal of stiffer cells by the spleen then relates to the extrapolation of our in vitro results to the situation in vivo. We have clarified the sentence in the main text to avoid confusion and we are looking forward to investigating hemorheological changes of in vivo parasitemia and thereby further assess the diagnostic value for patients with malaria.

It is unclear why the 2BP and PA treatments were performed. It would be helpful for the reader to clarify the motivation for these treatments as they relate to pf infection.

We realize that the 2BP and PA experiments had not been well motivated. Reduced membrane-cytoskeleton interactions have previously been correlated to elliptocytic RBCs and resistance to P.f. infection (Chishti et al., 1996). The characteristic biconcave morphology of RBCs can be chemically altered by the use of 2-bromo-palmitate (2BP), an efficient inhibitor of palmitoyl acyltransferases (PATs) (Lach et al., 2012). 2BP is commonly used to inhibit palmitoylation of proteins, but it should be also pointed out that it is an analogue of palmitic acid (PA) and as such will readily partition into the membranes. Since it is used at relatively high concentrations (100 µM), it might significantly change the biophysical properties of the membranes, where it accumulates. To discriminate between these two effects, we used palmitic acid, which similarly to 2BP will accumulate in the membranes, but will not inhibit protein palmitoylation. In this respect our data clearly point out that RBC protein palmitoylation and not accumulation of the “fatty acids” in the membranes, is critical for the infectivity of P. f. We have altered the main text to clarify the rationale for these experiments.

4) Subsection “MORE analysis of leukocytes” what is the timepoint of mechanical measurements of activated neutrophils published in the older reports? If measurements were performed at different time points, with different methods that deform cells on vastly different timescales, it does not seem to be a conflict.

Again, a very valuable comment. We have looked for the times when the effect of fMLP stimulation was determined in the older reports. In Worthen et al., 1989, treatment with fMLP was for 30 sec at different concentrations, but the time of measurement is not reported. As 30–80 cells were manually measured with a cell poker it has to be assumed that the duration of the experiment covered the entire timescale assessed in our experiments (0–60 min). Similarly, in Rosenbluth et al., 2008 neutrophils were exposed to 10 ng/mL fMLP for 5 min, but it is not reported how much later the measurements were taken. Both these studies reported a stiffening. The only other study we are aware of that reported a softening was Gossett et al., 2012. However, neutrophils were stimulated for 45 min so the initial stiffening the other reports (and we) find might have been missed. In addition, also the fMLP concentration of 10 µM used by Gossett at al. is rather high and non-physiological, and renders neutrophil incapable of depriming. These points, in addition to the fact that the different methods probe cells on vastly different timescales as the reviewers correctly state, the different modes of testing (see Ravetto et al., 2014), and the potentially different levels of pre-activation due to isolation of neutrophils from whole blood necessary for previous techniques (see also new results added in response to one of the minor comments below; Figure 1—figure supplement 3) render the perceived conflict, as stated in the original manuscript, insubstantial.

Neutrophils were less deformable after fMLP treatment, but became more deformable and larger at longer time points. Given that 'larger cells of identical stiffness should deform more in RTDC' (subsection “MORE analysis of malignant transformed blood cells”), how can these results show that fMLP cells, which are bigger, are less deformable? If larger cells are deforming more in RTDC (and when deformed by inertial flows as in Gossett et al.), then it is difficult to compare to previous methods.

A very good point. We had not been careful enough in only describing cells as more or less “deformed” and accidentally slipped in “deformable” here. As stated above, we can now actually calculate the apparent Young’s modulus of these cells and take the combined change of size and deformation into account. Indeed, there is first a significant increase in Young’s modulus for the first 15 min, which then drops back to slightly below the initial value (even though the difference to before is not significant). Considering this, and the previous point, we have now removed any mention of a seeming conflict between previous results, and instead describe in more detail the actual time-resolved mechanical changes taking place during the first hour of neutrophil activation, which has never been reported before.